# Reviews and syntheses: Potential and limitations of oceanic carbon dioxide storage via reactor-based accelerated weathering of limestone

Tom Huysmans<sup>1</sup>, Filip J. R. Meysman<sup>1</sup>, Sebastiaan J. van de Velde<sup>2,3,1</sup>

<sup>1</sup> Department of Biology, University of Antwerp, Wilrijk, 2610, Belgium

10 Correspondence to: Tom Huysmans (tom.huysmans@uantwerpen.be)





Abstract. To achieve climate stabilization, substantial emission reductions are needed. Emissions from industrial point sources can be reduced by applying CO<sub>2</sub> emission mitigation methods, which capture carbon dioxide (CO<sub>2</sub>) before it is released to the atmosphere. Accelerated weathering of limestone (AWL) is such a CO2 emission mitigation approach, in which calcium carbonate (CaCO<sub>3</sub>) is dissolved and CO<sub>2</sub> is stored as dissolved inorganic carbon in the ocean. At present, AWL technology remains at the pilot scale with no industrial implementation. Here, we review the proposed reactor designs for AWL, comparing them in terms of CO<sub>2</sub> capture efficiency, CaCO<sub>3</sub> dissolution efficiency, CO<sub>2</sub> sequestration efficiency, and water usage. For this, we represent AWL as a four step process: (i) CO<sub>2</sub> dissolution, (ii) CaCO<sub>3</sub> dissolution, (iii) alkalinization, and lastly (iv) re-equilibration. AWL application is generally characterized by a large water usage and the need for large reactor sizes. Unbuffered AWL approaches show substantial degassing of CO<sub>2</sub> back to the atmosphere after the process water is discharged. Buffered AWL approaches compensate the unreacted CO<sub>2</sub> by Ca(OH)<sub>2</sub> addition, which prevents degassing and hence substantially increases the CO2 sequestration efficiency. Critically however, buffered AWL requires a source of CO<sub>2</sub>-neutral Ca(OH)<sub>2</sub>, which is conventionally produced by calcination causing substantial CO<sub>2</sub> emissions. The need for process water can be reduced by increasing the CO<sub>2</sub> fraction of the gas stream or increasing its pressure. Further optimization of the size distribution of pulverized CaCO<sub>3</sub> particles could reduce the amount of Ca(OH)<sub>2</sub> needed to buffer the unreacted CO<sub>2</sub>. The anticipated CO<sub>2</sub> sequestration efficiency of buffered AWL is comparable with that projected for large-scale CCS in geological reservoirs.

<sup>&</sup>lt;sup>2</sup> Department of Marine Science, University of Otago, Dunedin, 9016, New-Zealand

<sup>&</sup>lt;sup>3</sup> National Institute of Water and Atmospheric Research, Wellington, 6022, New-Zealand

#### 30 1. Introduction

Atmospheric CO<sub>2</sub> levels have increased by ~50 % compared to preindustrial times and are higher than any period in the past two million years (Calvin et al., 2023). The 2015 Paris climate agreement aims to prevent global temperatures from rising more than 2 °C compared to preindustrial levels (Sanderson et al., 2016). To this end, climate policies are focused on the reduction of greenhouse gas (GHG) emissions, combining a reduced usage of fossil fuels with the development of CO<sub>2</sub> emission mitigation technologies, which capture carbon dioxide (CO<sub>2</sub>) before it is released to the atmosphere. To provide a timely and meaningful contribution to climate mitigation, these CO<sub>2</sub> emission mitigation technologies need to be implemented at the Gigaton scale within the next decade, which requires a strong acceleration of their development (United Nations Environment Programme, 2024).

Industrial point-source CO<sub>2</sub> emissions from waste gas streams can be mitigated by geochemical-based processes in which CO<sub>2</sub> is reacted with solid carbonate or silicate rocks in the presence of water, which aims to enhance the natural weathering process of carbonate and silicate rocks (Rau and Caldeira, 1999; Renforth and Kruger, 2013; Caserini et al., 2021). This targeted weathering process can take place *in situ*, in which CO<sub>2</sub> is first captured from the flue gas and then injected into suitable silicate rock formations (basalts and ultramafic rocks). The CO<sub>2</sub> is then trapped by a carbonation reaction with the ambient silicate rock, thus ensuring a permanent, geological storage (Matter and Kelemen, 2009; Romanov et al., 2015; Gadikota, 2021; Cao et al., 2024). However, there are certain geomechanical risks associated with geological storage of CO<sub>2</sub>, such as CO<sub>2</sub> leakage, induced seismicity, the loss of well integrity, and surface uplift (Song et al., 2023). Moreover, suitable rock formations for storage are not always in close proximity to the CO<sub>2</sub>-emitting installations, thus requiring compression/liquefaction and transport of CO<sub>2</sub>.

Alternatively, the chemical weathering can also be executed under controlled conditions in a land-based reactor, close to the industrial point source. Mitigation of CO<sub>2</sub> emissions via such reactor-based methods can follow two main approaches, depending on whether silicates are used as feedstock material (usually referred to a "ex-situ mineral carbonation" technologies; Romanov et al., 2015; Gadikota, 2021, or "mineralization"; Campbell et al., 2022) or whether carbonates are used as weathering substrates (referred to a as "accelerated weathering of limestone"; Rau and Caldeira, 1999). In ex-situ mineral carbonation (ESMC), a finely-ground silicate mineral (e.g. olivine Mg<sub>2</sub>SiO<sub>4</sub>) is fed into a reactor, where it reacts at elevated temperature and pressure with CO<sub>2</sub> from a flue gas to eventually form stable carbonates (e.g. magnesite Mg<sub>2</sub>SiO<sub>4</sub>) - see recent reviews (Snæbjörnsdóttir et al., 2020; Veetil and Hitch, 2020; Thonemann et al., 2022). Alternatively, during the accelerated weathering of limestone (AWL), CO<sub>2</sub> is stripped from the flue gas using a mixture of seawater and limestone (Rau and Caldeira, 1999; Renforth and Henderson, 2017), and the resulting effluent is discharged into the sea.

The main difference between the two approaches is that ESMC stores  $CO_2$  in a mineral form, whereas AWL stores  $CO_2$  in dissolved form in the ocean. As such, AWL bears similarities with so-called ocean alkalinization approaches, which target the deliberate removal of  $CO_2$  directly from the atmosphere, by increasing the alkalinity  $(A_T)$  of the surface ocean (Kheshgi, 1995; Meysman and Montserrat, 2017; Renforth and Henderson, 2017). The natural weathering of silicate and carbonate rocks generates  $A_T$  (Berner and Berner, 2004), which is transported by rivers to the ocean. Increasing seawater  $A_T$ , which is defined as the excess of base (proton acceptors) over acid (proton donors) (Dickson, 1981; Zeebe and Wolf-Gladrow, 2001), shifts the carbonate equilibrium away from dissolved  $CO_2$  towards bicarbonate (HCO<sub>3</sub>-) and carbonate ( $CO_3$ -) ions. As a result, more atmospheric  $CO_2$  can

be stored in seawater as dissolved inorganic carbon (DIC; defined as the sum of the aqueous [CO<sub>2</sub>], [HCO<sub>3</sub><sup>-</sup>], and [CO<sub>3</sub><sup>2</sup>-] concentrations; Zeebe and Wolf-Gladrow, 2001). This natural process of ocean alkalinization, induced by the chemical weathering of rocks, has regulated atmospheric CO<sub>2</sub> and stabilized the climate over geological time scales (Berner et al., 1983). The process of AWL aims to mimic the natural process of carbonate weathering in a reactor, but in an accelerated fashion. Here, we review the potential of AWL as a CO<sub>2</sub> emission mitigation approach, including its intricacies and possible bottlenecks. To this end, we describe AWL thermodynamically as a four step process, thus providing a model framework that allows to calculate the efficiency of the different steps as well as the overall CO<sub>2</sub> sequestration potential. We then review the different reactor designs that have been proposed for the AWL process in recent years and evaluate their efficiency and potential in terms of CO<sub>2</sub> emission mitigation capacity.

#### 2. The theoretical principle of AWL

#### 2.1. AWL as a four-step process







The concept of AWL was first proposed more than two decades ago by Rau and Caldeira (1999). It provides a geochemistry-based method for CO2 emissions mitigation in which the aqueous reaction of carbonate minerals (e.g. CaCO<sub>3</sub>) with CO<sub>2</sub> is enhanced due to the elevated concentration of CO<sub>2</sub> as typically encountered in waste gas streams of industrial combustion processes (Rau and Caldeira, 1999). Finely ground carbonate (e.g., calcite, aragonite, dolomite or magnesite) and a suitable stream of process water are brought into direct contact with the flue gasses from a CO<sub>2</sub>-intensive industrial source, such as a coal-fired power plant or a cement factory (Fig. 1). In general, the process of AWL can be described as consisting out of four different steps (Fig. 1): (i) CO<sub>2</sub> uptake: the process water comes into contact with the flue gas, which has a much higher partial pressure of CO<sub>2</sub> than the ambient atmosphere (typically pCO<sub>2</sub>  $\approx 0.15$  atm). This leads to dissolution of CO<sub>2</sub> in the process water, thus increasing the DIC, and lowering the pH and calcite saturation state ( $\Omega_{calc}$ ), while keeping  $A_T$  constant; (ii) CaCO<sub>3</sub> **dissolution**: The reduced  $\Omega_{calc}$  of the process water stimulates the dissolution of carbonate particles and increases both the DIC and A<sub>T</sub> of the process water. Subsequently, there are two options. In the case of 'buffered AWL" (Caserini et al., 2021), there is an additional (iii) alkalinization step before re-equilibration to avoid the degassing of CO<sub>2</sub>. Additional A<sub>T</sub> is added to the process water (e,g. by addition of slaked lime, Ca(OH)<sub>2</sub>) until the excess CO<sub>2</sub> is fully buffered. After discharge into the surface ocean, there is no longer any CO<sub>2</sub> transfer to the atmosphere. In the case of 'unbuffered AWL", there is the (iv) re-equilibration step: The process water is discharged into the sea without any further treatment after which it re-equilibrates with the atmosphere at the lower pCO<sub>2</sub> (pCO<sub>2</sub>  $\approx$ 0.00042 atm) and the excess CO<sub>2</sub> (i.e., the part of DIC not stabilized by the increased A<sub>T</sub>) will degas back to the atmosphere.

Below we discuss each step in more detail. During the whole AWL process, the process water goes through four consecutive states, each characterized by a specific set of A<sub>T</sub>, DIC, pCO<sub>2</sub>, and pH values. These states are: (1) the ambient process water that is used as intake, (2) the process water with elevated DIC after CO<sub>2</sub> uptake, (3) the process water enriched in A<sub>T</sub> and DIC after CaCO<sub>3</sub> dissolution, (4a) the unbuffered or (4b) buffered process water after discharge into the surface ocean.

Figure 1. The process of accelerated weathering of limestone can be described by four different steps: (i)  $CO_2$  uptake:  $CO_2$  from the flue gas comes in contact with the process water and  $CO_2$  dissolves into the process water, (ii)  $CaCO_3$  dissolution: Aqueous  $CO_2$  reacts with  $CaCO_3$  particles and generates  $A_T$  in the form of  $HCO_3$ , which is stimulated by the reduced  $\Omega_{cak}$ , (iii) the alkalinization step (in buffered AWL): Additional  $A_T$  is added to the process water (e,g. by slaked lime addition), until the excess  $CO_2$  is fully buffered, and (iv) the re-equilibration step: Upon re-exposure to atmospheric conditions, aqueous  $CO_2$  which is not stabilized by the increased  $A_T$  will degas back to the atmosphere. The black lines indicate the gas flows and the blue lines indicate the process water flows.







Table 1 shows the values for pCO<sub>2</sub>,  $A_T$ , DIC, pH, and  $\Omega_{calc}$  in each of the four states for a representative case, which is based on data reported from a two-step bench-top reactor consisting of a separate gas-liquid and liquidsolid reactor (Chou et al., 2015, reactor design as further discussed below). The CO<sub>2</sub> concentration of the gas stream was 15%, while the pCO<sub>2</sub> of the atmosphere is fixed at 420 ppm. The A<sub>T</sub> and DIC values at the inlet and outlet of the reactor are based on measured values (Table 1 in Chou et al., 2015). The remaining variables are calculated using the CRAN: AquaEnv package for the thermodynamic equilibria of acid-base systems in seawater (Hofmann et al., 2010). We assume full re-equilibration with the atmosphere (unbuffered AWL) or full buffering with slaked lime (Ca(OH)<sub>2</sub>) upon discharge into the sea (buffered AWL). This condition of full re-equilibration requires consideration. In the well-mixed coastal zone, air-sea CO<sub>2</sub> exchange takes place on a time-scale of several weeks up to a year (Jones et al., 2014; He and Tyka, 2023; Geerts et al., 2025). When the surface residence time of the discharged process water is shorter than the air-sea CO<sub>2</sub> equilibration timescale, some of the dissolved CO<sub>2</sub> unbuffered by the A<sub>T</sub> increase in the AWL reactor can move to deeper layers and so full re-equilibration will not be reached (Jones et al., 2014; He and Tyka, 2023). Likewise, when the process water is discharged below the stratification layer or directly in the deeper ocean, full re-equilibration will also be prevented (Jones et al., 2014; He and Tyka, 2023). In both the cases, the CO<sub>2</sub> sequestration is increased. Therefore, assuming full reequilibration represents a conservative lower bound for the CO<sub>2</sub> sequestration during AWL.

The transition through the different consecutive states is depicted in the thermodynamic diagrams of Fig. 2, which each depict the gas phase pCO<sub>2</sub> versus the process water A<sub>T</sub>, but with different isolines (DIC, pH, and  $\Omega_{\rm cale}$ ). Changes in the chemical conditions of the inlet process water, the water/gas flow rate ( $Q_{\rm water}/Q_{\rm gas}$ ), the pCO<sub>2</sub> of the gas stream, or the reactor setup will modify the modelled parameters presented in Table 1 and Figure 2.

Table 1. Theoretical values for alkalinity ( $A_T$ ), dissolved inorganic carbon (DIC), pH, and calcite saturation state ( $\Omega_{calc}$ ) in the four consecutive states of the example AWL reactor: (1) the process water that is used as intake (the process water was collected from an offshore station near the Hoping power plant and the inlet and outlet of the cooling water drainage of the Hoping power plant (Chou et al., 2015)), (2) the process water with elevated DIC after  $CO_2$  uptake, (3) the process water enriched in  $A_T$  and DIC after  $CaCO_3$  dissolution, (4a) the unbuffered or (4b) buffered process water upon discharge.  $\Delta$ DICseq is the DIC that is added to the process water due to dissolution from the gas stream and  $\Delta$ DICcarb is the DIC added through the dissolution of  $CaCO_3$  in the reactor. The pCO<sub>2</sub>,  $A_T$  and DIC values (indicated by #) are based on values measured in a two-step AWL bench-top reactor (Chou et al., 2015). The values of  $A_T$ , DIC, pH, and  $\Omega$ calc (indicated with \*) are calculated using CRAN:AquaEnv (Hofmann et al., 2010) for seawater at a temperature of 15 °C and salinity of 35.





| State | pCO <sub>2</sub> | $\mathbf{A}_{T}$ | DIC         | $\Delta DIC_{seq}$ | <b>ADIC</b> carb | pН         | $\Omega_{ m calc}$ |
|-------|------------------|------------------|-------------|--------------------|------------------|------------|--------------------|
|       | (atm)            | (mM)             | (mM)        | (mM)               | (mM)             | (-)        | (-)                |
| (1)   | 0.000420         | $2.26^{\#}$      | 2.13#       | 0                  | 0                | $7.93^{*}$ | $2.50^{*}$         |
| (2)   | 0.15 #           | 2.26             | $2.96^{*}$  | 0.83               | 0                | $6.52^{*}$ | $0.110^{*}$        |
| (3)   | 0.15             | $2.64^{\#}$      | $3.15^{\#}$ | 0.83               | 0.19             | $6.72^{*}$ | $0.203^{*}$        |
| (4a)  | 0.000420         | 2.64             | $2.38^{*}$  | 0.06               | 0.19             | $8.16^{*}$ | $4.62^{*}$         |
| (4b)  | 0.000420         | $3.56^{*}$       | $3.15^{*}$  | 0.83               | 0.19             | $8.27^{*}$ | $7.74^{*}$         |

During step (i), the A<sub>T</sub> remains invariant between state (1) and state (2) (vertical trajectory in Fig. 2). The high CO<sub>2</sub> concentration in the flue gas drives the dissolution of CO<sub>2</sub> into the water phase, which increases the DIC of the process water (Fig. 2a), lowers its pH (Fig. 2b), and drastically lowers the Ω<sub>calc</sub> (Fig. 2c; Table 1). As a result, the dissolution of CaCO<sub>3</sub> in step (ii) becomes thermodynamically favorable, and because of the strong disequilibrium, the dissolution rate is increased (Berner and Morse, 1974; Morse et al., 2007). Note that the effluent at state 3 in the example two-step reactor is not in equilibrium with respect to CaCO<sub>3</sub> dissolution (Ω<sub>calc</sub> < 1, Table 1). This indicates that the effectiveness of CaCO<sub>3</sub> dissolution in the reactor design of Chou et al. (2015) could still be improved (e.g. by implementing a longer residence time). The dissolution of CaCO<sub>3</sub> can be described by the reaction:

$$CO_2 + H_2O + CaCO_3 \rightarrow Ca^{2+} + 2HCO_3^-$$
 (1)

Because the input of A<sub>T</sub> from CaCO<sub>3</sub> dissolution is twice that of DIC, the carbonate equilibrium in the process water is shifted away from CO<sub>2</sub> towards HCO<sub>3</sub><sup>-</sup> and CO<sub>3</sub><sup>2-</sup> (Eq. 2), which slightly increases the pH and calcite saturation state (Fig. 2; Table 1).

$$H_2O + CO_2 \leftrightarrow HCO_3^- + H^+ \leftrightarrow CO_3^{2-} + H^+$$
 (2)

In the unbuffered AWL scenario, the effluent water of the reactor is simply discharged in the marine environment and is re-exposed to the atmosphere. We can model this as a re-equilibration of the process water with the ambient atmospheric pCO<sub>2</sub>, step (iv), which will induce an outgassing of excess dissolved CO<sub>2</sub>. The release of CO<sub>2</sub> from the effluent results in a marked decrease of DIC and a concomitant increase in pH and  $\Omega_{calc}$  (Fig. 2; Table 1).

Two assumptions are worth noting. In our scheme, we assumed that the effluent process water first equilibrates with the ambient atmosphere, before it is mixed with the surrounding seawater. In reality, the process water will be mixed first with ambient seawater. However, one can easily show that equilibration followed by mixing, provides the same  $CO_2$  transfer as mixing followed by equilibration. Secondly, the calcite saturation state of the solution after degassing is larger than one. Such a supersaturated solution could (at least in theory) induce the reprecipitation of  $CaCO_3$  within the marine environment with a resulting loss of  $A_T$ . Still, the abiotic precipitation of  $CaCO_3$  in seawater typically requires a highly supersaturated solution ( $\Omega_{calc} > 18$ ) (Morse and He, 1993).

Therefore abiotic CaCO<sub>3</sub> formation is unfavorable from supersaturated seawater and rare under natural conditions (Mucci et al., 1989; Moras et al., 2022; Hartmann et al., 2023). Accordingly, we assume that no carbonate precipitation takes place after the discharge of the process water.





In the buffered AWL scenario,  $Ca(OH)_2$  is added to the process water before its discharge into the marine environment (Caserini et al., 2021). During this step, all the unreacted  $CO_2$  is buffered, which hence prevents any loss of DIC (Fig 2a), increases  $A_T$  and pH, and also substantially increases  $\Omega_{calc} \sim 8$  (Fig. 2b-c). While the abiotic precipitation of  $CaCO_3$  is kinetically inhibited under such high  $\Omega_{calc}$  values (see above), its risk could be further reduced by: 1) discharging the process water where rapid mixing and dilution occurs, 2) mixing the process water with deeper and colder waters, which increases the solubility of  $CaCO_3$ , or 3) injection of the process water at a depth below the calcite compensation depth (Kirchner et al., 2020a).

Figure 2. Changes in carbonate chemistry for the four different steps during AWL: (i)  $CO_2$  uptake:  $CO_2$  gas from the flue gas comes in contact with the process water and  $CO_2$  dissolves into the process water, (ii)  $CaCO_3$  dissolution: Aqueous  $CO_2$  reacts with  $CaCO_3$  particles and generates  $A_T$  in the form of  $HCO_3$ , which is stimulated by the reduced saturation state, (iii) the alkalinization step (in buffered AWL): Additional  $A_T$  is added to the process water (e,g. by  $Ca(OH)_2$  addition), until the excess  $CO_2$  is fully buffered and (iv) the re-equilibration step: Upon re-exposure to atmospheric conditions, aqueous  $CO_2$  which is not stabilized by the increased  $A_T$  will degas back to the atmosphere.  $pCO_2$  (atm) in function of  $A_T$  (mmol kg<sup>-1</sup>) with isolines for a) DIC, b) pH and c)  $\Omega_{calc}$ . The DIC concentration in the process water has increased over the course of the three consecutive steps indicating a capture of  $CO_2$ .

### 2.2. CO<sub>2</sub> sequestration during CaCO<sub>3</sub> dissolution and Ca(OH)<sub>2</sub> buffering

Overall, the A<sub>T</sub> increase following CaCO<sub>3</sub> dissolution leads to the sequestration of CO<sub>2</sub> from the flue-gas in the form of DIC in the seawater (Rau and Caldeira, 1999; Caldeira and Rau, 2000; Rau et al., 2007; Rau, 2011). As can be seen from Table 1, the final DIC (2.38 mM in the unbuffered case; 3.15 mM in the buffered case) is higher than in the intake water (2.13 mM). However, only part of this DIC increase is due to CO<sub>2</sub> sequestration from the

flue gas, as part of the additional DIC also originates from CaCO<sub>3</sub> dissolution. To separate the different effects that contribute to CO<sub>2</sub> sequestration, the DIC increase can be decomposed as:



$$\Delta DIC_{total} = DIC_{final} - DIC_{inlet} = \Delta DIC_{seq}^{unbuf} + \Delta DIC_{seq}^{buf} + \Delta DIC_{carb}$$
(3)

 $DIC_{inlet}$  is the DIC value measured in the process water at the inlet,  $\Delta DIC_{carb}$  denotes the DIC that originates from CaCO<sub>3</sub> during dissolution,  $\Delta DIC_{seq}^{unbuf}$  represents the DIC in the process water that originates from net CO<sub>2</sub> sequestration from the flue gas in the reactor and  $\Delta DIC_{seq}^{buf}$  represents the DIC that is retained (i.e. prevented from efflux to the atmosphere) due to the Ca(OH)<sub>2</sub> buffering of the effluent (in the unbuffered scenario  $\Delta DIC_{seq}^{buf} = 0$ ). In a similar fashion, the final A<sub>T</sub> value is the result of A<sub>T</sub> addition during CaCO<sub>3</sub> dissolution and the A<sub>T</sub> that is added during buffering with Ca(OH)<sub>2</sub> in the case of buffered AWL.

$$\Delta A_{T,total} \equiv A_{T,final} - A_{T,inlet} = \Delta A_{T,carb} + \Delta A_{T,buf}$$
(4)

From this, the net CO<sub>2</sub> sequestration is obtained by subtraction of the DIC that originates from CaCO<sub>3</sub> dissolution:

$$\Delta DIC_{seq} = \Delta DIC_{seq}^{unbuf} + \Delta DIC_{seq}^{buf} = \Delta DIC_{total} - \Delta DIC_{carb}$$
(5)

In practical AWL applications, the  $\Delta$  quantities can be determined by measuring DIC and  $A_T$  at the inlet and outlet of the AWL reactor (i.e., before the buffering step), complemented by thermodynamic calculations (see Table 1). The DIC and  $A_T$  increase due to CaCO<sub>3</sub> dissolution can be directly inferred from the stoichiometry of the CaCO<sub>3</sub> dissolution reaction Eq. (1):

$$\Delta A_{T,carb} = A_{T,outlet} - A_{T,inlet}, \quad \Delta DIC_{carb} = \frac{A_{T,outlet} - A_{T,inlet}}{2} = \frac{1}{2} \Delta A_{T,carb}$$
 (6)

For every mole of  $CaCO_3$  that dissolves, two moles of  $A_T$  are formed and one extra mole of DIC is generated from the  $CaCO_3$ . Therefore, the amount of DIC generated from  $CaCO_3$  dissolution is half the amount of  $A_T$  increase between the inlet and outlet of the reactor.

In AWL applications, the critical quantity is the overall DIC increase resulting from net  $CO_2$  sequestration, i.e.,  $\Delta DIC_{seq}$ . Here we need to make a distinction between the buffered and unbuffered scenario. In the unbuffered scenario, one calculates the DIC and  $A_T$  values after re-equilibration of the process water with the atmosphere.

$$A_{T, final} = A_{T, outlet} = A_{T, inlet} + \Delta A_{T, carb} \tag{7}$$

$$DIC_{final} = f(A_{T,final}, pCO_{2,atm}) \approx DIC_{inlet} + \left(\frac{\partial DIC}{\partial A_T}\right)_{pCO_{2,atm}} \Delta A_{T,carb}$$
 (8)

The  $A_T$  concentration does not change during re-equilibration (remains same as the outlet), while the final DIC value can be calculated from this  $A_T$  concentration and the atmospheric pCO<sub>2</sub> based on thermodynamic relations of seawater carbonate chemistry (assuming full equilibration with the atmosphere). The approximation in Eq. (9) uses the thermodynamic buffer factor  $\gamma = (\partial DIC/\partial A_T)_{pCO_{2,amm}}$ , which specifies the increase in seawater DIC taken due to CO<sub>2</sub> uptake from the atmosphere given a certain addition of  $A_T$  (Zeebe and Wolf-Gladrow, 2001). This buffer factor is calculated at the atmospheric pCO<sub>2</sub> and ambient seawater concentrations (i.e., inlet conditions), which serves as a reasonable approximation, since the outlet water will be quickly mixed with ambient seawater. Accordingly, in the unbuffered scenario, the total amount of CO<sub>2</sub> sequestered becomes:

$$\Delta DIC_{sea}^{unbuf} = DIC_{final} - DIC_{inlet} - \Delta DIC_{carb} = (2\gamma - 1)\Delta DIC_{carb}$$
(9)

The amount of CO<sub>2</sub> that is lost via outgassing upon re-equilibration can be calculated as:

$$\Delta DIC_{outvas} = DIC_{outlet} - DIC_{final} \tag{10}$$

Alternatively, in the case of buffered AWL, one adds additional A<sub>T</sub> to the effluent water, until equilibrium is reached with the ambient atmosphere, and so no CO<sub>2</sub> will be outgassed to the atmosphere. The final state is calculated as:

$$DIC_{final} = DIC_{outlet}$$
(11)

$$A_{T,final} = f(DIC_{outlet}, pCO_2^{atm}) \approx A_{T,inlet} + \frac{1}{\gamma} \left(DIC_{outlet} - DIC_{intlet}\right)$$
(12)

The final A<sub>T</sub> value can again be calculated from thermodynamic relations of seawater carbonate chemistry. The amount of A<sub>T</sub> that needs to be supplied by Ca(OH)<sub>2</sub> addition to achieve "full buffering" is given by:

$$\Delta A_{T,buffer} = A_{T,final} - A_{T,inlet} - \Delta A_{T,carb} = \frac{1}{\gamma} \left( DIC_{outlet} - DIC_{intlet} \right) - \Delta A_{T,carb}$$
(13)

Accordingly, in the buffered scenario, the total amount of CO<sub>2</sub> sequestered can be calculated as:

$$\Delta DIC_{seq} = DIC_{outlet} - DIC_{inlet} - \Delta DIC_{carb}$$
(14)

240 The amount of CO<sub>2</sub> sequestration that is generated by buffering can be calculated as

$$\Delta DIC_{sea}^{buf} = \Delta DIC_{sea} - \Delta DIC_{sea}^{unbuf} \tag{15}$$

In our example (Table 1), the total DIC increase in the equilibrated effluent water amounts to  $\Delta DIC_{total} = 0.25$  mM in the unbuffered case, of which 76 % (0.19 mM) originates from CaCO<sub>3</sub> dissolution and 24% (0.06 mM) is due to CO<sub>2</sub> sequestration from the flue gas. In the buffered case, the DIC increase in the buffered discharge water amounts to  $\Delta DIC_{total} = 1.02$  mM of which 19% (0.19 mM) originates from CaCO<sub>3</sub> dissolution, 6% (0.06 mM) is due to unbuffered CO<sub>2</sub> sequestration and 75% (0.77 mM) results from additional (buffered) CO<sub>2</sub> sequestration via dissolution of Ca(OH)<sub>2</sub>. This illustrates how in the unbuffered scenario, a large fraction of the CO<sub>2</sub> initially sequestered from the flue gas escapes back to the atmosphere upon release of the reactor water into the ocean.

The operation and performance of an AWL reactor can be quantified by introducing a number of efficiency factors, which can be calculated from the  $\Delta DIC_{seq}$  and  $\Delta DIC_{carb}$  values defined above (and hence from A<sub>T</sub> and DIC values measured at the inlet and outlet of the reactor). These efficiency factors can again be linked to the different steps in the AWL process (as in Figure 2), and will allow us to compare the efficiency of different reactor designs. We now first introduce these efficiency factors formally.

## 2.3. CO<sub>2</sub> sequestration efficiency and water usage

245

250

The key target of the AWL reactor is to remove  $CO_2$  from the gas stream and store this permanently as DIC in the surface ocean. This performance is quantified by the  $CO_2$  sequestration efficiency ( $\eta_{seq}$ ), which is defined as the fraction of  $CO_2$  sequestered from the gas stream, accounting for re-equilibration with the atmosphere and associated  $CO_2$  degassing and buffering:

$$\eta_{seq} = \frac{\Delta DIC_{seq}RT}{\left(pCO_{2,sqs} - pCO_{2,atm}\right)} \frac{Q_{w}}{Q_{g}} \tag{16}$$

In this, the reactor is fed with a gas stream Q<sub>g</sub> (m³ s⁻¹) at a certain CO<sub>2</sub> partial pressure ( $pCO_{2,gas}$ ), and uses a process water stream Q<sub>w</sub> (m³ s⁻¹) which is characterized by  $DIC_{inlet}$  and  $A_{T,inlet}$ . R is the ideal gas constant (L atm mol⁻¹ K⁻¹) and T is the temperature of the gas stream (K). The maximum CO<sub>2</sub> sequestration efficiency is achieved when upon exit, the process water is in full equilibrium with the flue gas and all the dissolved CO<sub>2</sub> in the process water is suitably buffered by CaCO<sub>3</sub> dissolution in the AWL reactor and/or additional Ca(OH)<sub>2</sub> buffering, i.e.,  $\Delta DIC_{seq}^{max} = DIC_{eq} - DIC_{inlet}.$ 

$$\eta_{seq}^{\max} = \frac{\left(DIC_{eq} - DIC_{inlet}\right)RT}{\left(pCO_{2,gas} - pCO_{2,alm}\right)} \frac{Q_{w}}{Q_{g}} \tag{17}$$

The equilibrium value,  $DIC_{eq}\left(A_T^{inlet}, pCO_{2,gas}, T, S\right)$  can be calculated from carbonate chemistry as a function of the  $A_T$  of the inlet water and the pCO<sub>2</sub> of the gas stream. From this, the minimum water to gas flow ratio ( $Q_{w,\min}/Q_g$ ) that is required to achieve 100% CO<sub>2</sub> sequestration efficiency ( $\eta_{seq}^{\max}=1$ ) can be calculated as:

$$\frac{Q_{w,\min}}{Q_g} = \frac{(pCO_{2,gas} - pCO_{2,atm})}{RT(DIC_{eq} - DIC_{inlet})}$$
(18)

In our example reactor, this  $Q_{w,\min}/Q_g$  amounts to 0.76 (Table 2). A water efficiency factor ( $W_{eff}$ ) can be defined as actual water consumption of the reactor over the minimum required  $Q_w$  to achieve maximum sequestration.

$$W_{eff} = \frac{Q_{w}}{Q_{winin}} \tag{19}$$

If  $W_{eff}$  is smaller than 1, the  $Q_w$  is not sufficient to dissolve all the CO<sub>2</sub> in the gas stream down to atmospheric pCO<sub>2</sub> and so the sequestration efficiency is limited by the  $Q_w$  ( $\eta_{seq}^{max} < 1$ ). If the  $W_{eff}$  is larger than 1, more water is used than is strictly required. In our example reactor, the maximum CO<sub>2</sub> uptake efficiency is 100% and  $W_{eff} = 3.2$  (Table 2). The volume of process water (m³) that is used to capture one tonne of CO<sub>2</sub> can be calculated from Eq. (17) as:

$$V_{water} = \frac{1}{\Delta DIC_{seq}} \frac{10^{-6}}{M_{CO_2}} \tag{20}$$

In this,  $M_{CO2}$  is the molar mass of  $CO_2$  (44.01 g mol<sup>-1</sup>) and  $10^{-6}$  is used to convert g to tonne (1 g =  $10^{-6}$  tonne), while  $\Delta DIC_{seq}$  is expressed in mol per unit of volume. In our reactor example, 150.000 m<sup>3</sup> of process water is used to capture 1 tonne of  $CO_2$ , thus illustrating the large water footprint of AWL.

### 2.4. CO<sub>2</sub> dissolution efficiency and CaCO<sub>3</sub> dissolution efficiency

285

In reality, the maximum  $CO_2$  sequestration efficiency will not be reached, due to several forms of inefficiency. In the first step, there might be incomplete dissolution of  $CO_2$  in the inlet water from the flue gas stream. To account for this, the  $CO_2$  dissolution efficiency is defined as the amount of  $CO_2$  that is effectively removed from the gas stream versus its theoretical maximum

$$\varepsilon_{CO_2} = \frac{DIC_{outlet} - DIC_{inlet} - \Delta DIC_{carb}}{DIC_{eq} - DIC_{inlet}}$$
(21)

The maximum  $CO_2$  dissolution efficiency of 100% is reached when  $DIC_{outlet} = DIC_{eq} + \Delta DIC_{carb}$ . The  $CO_2$  uptake efficiency is defined as the relative amount of  $CO_2$  that is stripped from the incoming gas stream (irrespective of whether it is eventually sequestered or not – see below)

$$\eta_{uptake} = \varepsilon_{CO_2} \eta_{seq}^{\max} \tag{22}$$

As can be seen, the CO<sub>2</sub> uptake efficiency is critically dependent on the CO<sub>2</sub> dissolution efficiency  $\varepsilon_{CO_2}$  as well as the  $Q_w/Q_g$  ratio at which the reactor operates (which defines  $\eta_{seq}^{max}$ ). In the example reactor, the CO<sub>2</sub> uptake efficiency ( $\eta_{uptake}$ ) becomes 33%, implying that only one third of the CO<sub>2</sub> is removed from the gas stream.

In a second step, the dissolution of  $CaCO_3$  in the AWL reactor targets the neutralization the dissolved  $CO_2$  by its conversion to  $HCO_3^-$  via reaction Eq. (1). The **CaCO<sub>3</sub> dissolution efficiency** is defined as the percentage of the dissolved  $CO_2$  within the reactor that has reacted with  $CaCO_3$ .

$$\varepsilon_{CaCO_3} = \frac{\Delta DIC_{carb}}{DIC_{outlet} - \Delta DIC_{carb} - DIC_{inlet}}$$
(23)

The maximum  $CaCO_3$  dissolution efficiency is reached when the DIC released during  $CaCO_3$  dissolution matches the amount of  $CO_2$  extracted from the gas phase, i.e.,  $\Delta DIC_{carb} = 1/2(DIC_{outlet} - DIC_{inlet})$ . In the example reactor, the  $CaCO_3$  dissolution efficiency is 22%, implying that only a part of the  $CO_2$  extracted from the gas stream is buffered by  $CaCO_3$  dissolution.

#### 2.5. Outgassing and buffering effects

295

315

The outgassing effect  $\varepsilon_{outgas}$  is defined as the amount of CO<sub>2</sub> sequestered in the unbuffered scenario relative to the amount of CO<sub>2</sub> that has reacted with CaCO<sub>3</sub>:

$$\varepsilon_{outgas} = \frac{\Delta DIC_{seq}^{umbuf}}{\Delta DIC_{carb}} = (2\gamma - 1) \tag{24}$$

As shown in Eq. (9), the outgassing effect  $\varepsilon_{outgas}$  is directly proportional to the thermodynamic buffer factor  $\gamma$ , which is always smaller than 1 and so  $\varepsilon_{outgas} < 1$ . Finally, the buffering effect is defined as:

$$\varepsilon_{buffer} = \frac{\Delta DIC_{seq}^{buf}}{\Delta DIC_{seq}^{unbuf}} = \frac{\Delta DIC_{seq}}{\Delta DIC_{seq}^{unbuf}} - 1 \tag{25}$$

Based on the factors introduced above, the effective CO<sub>2</sub> sequestration efficiency thus becomes:

$$\eta_{seq} = \frac{\Delta DIC_{seq}RT}{\left(pCO_{2,gas} - pCO_{2,atm}\right)} \frac{Q_{w}}{Q_{g}} = \varepsilon_{CO_{2}} \varepsilon_{CaCO_{3}} \left(2\gamma - 1\right) \left[1 + \varepsilon_{buffer}\right] \eta_{seq}^{max}$$
(26)

As apparent, the fact that the efficiencies  $\varepsilon_{CO_2}$ ,  $\varepsilon_{CaCO_3}$  and  $\gamma$  are lower than 1 decreases the CO<sub>2</sub> sequestration efficiency below its maximal attainable value. When there is no buffering ( $\varepsilon_{buffer} = 0$ ) then  $\eta_{seq} = \varepsilon_{CO_2} \varepsilon_{CaCO_3} (2\gamma - 1) \eta_{seq}^{max}$ . In contrast, when there is maximum buffering, the relation  $\eta_{seq} = \varepsilon_{CO_2} \eta_{seq}^{max} = \eta_{uprake}$ 

holds, and so the  $CO_2$  uptake efficiency is always the same as the  $CO_2$  sequestration efficiency. In this scenario, the buffering compensates entirely for incomplete  $CaCO_3$  dissolution and prevents outgassing (i.e.,  $\varepsilon_{buffer} = \left[1 - \varepsilon_{CaCO_3} \left(2\gamma - 1\right)\right] / \left[\varepsilon_{CaCO_3} \left(2\gamma - 1\right)\right]$ ). In our example reactor, the unbuffered  $CO_2$  sequestration efficiency is only 6% (see Table 2), while the buffered  $CO_2$  sequestration efficiency (or equally, the  $CO_2$  uptake efficiency) amounts to 33%, thus indicating that a large part of the  $CO_2$  initially gained will be lost by outgassing upon re-equilibration.

#### 3. Different reactor designs for AWL






Over the past decades, several reactor designs have been proposed for AWL. Some have remained at a conceptual model stage, while others have been tested in bench-top or pilot scale operations (Table 2). As such, the technological readiness level is still limited and restricted to pilot scale applications (Chou et al., 2015; Kirchner et al., 2020b). In this section, we will compare four different reactor designs: a one-step reactor (Rau, 2011; Chou et al., 2015), a two-step reactor (Chou et al., 2015), a slurry reactor (Kirchner et al., 2020b) and a buffered AWL reactor (Caserini et al., 2021). The operational conditions and process efficiencies of these reactor designs are summarized in Table 2. The presented operational conditions are given for specific example reactor setups (benchtop (Chou et al., 2015) or pilot plant (Kirchner et al., 2020b)) or conceptual designs (Caserini et al., 2021) and the process efficiencies are calculated based on published data for a specific operational condition. Changes in reactor design or operational conditions will change these calculated efficiencies.

Table 2: Operational and process conditions for an example of a one- and two-step reactor (Chou et al., 2015), a slurry reactor (Kirchner et al., 2020b) and a buffered AWL reactor (Caserini et al., 2021). \* = after the dissolution reactor, \*\* = after the buffering reactor as no degassing takes place. When water and/or gas flow rates are not specified, no  $CO_2$  uptake or sequestration efficiency can be calculated, as was the case for Two-Step and buffered AWL.

|                        |                                                                                               | 0         | Т         | C1          | D - CC - 1 A XVI |
|------------------------|-----------------------------------------------------------------------------------------------|-----------|-----------|-------------|------------------|
|                        |                                                                                               | One-step  | Two-step  | Slurry      | Buffered AWL     |
| Operational conditions | Operational stage                                                                             | Bench-top | Bench-top | Pilot       | Conceptual       |
|                        | pCO <sub>2</sub> of the gas stream (atm)                                                      | 0.15      | 0.15      | 0.10 - 0.12 | 0.28             |
|                        | water/gas flow ratio (v/v)                                                                    | 3.5       | 2.6       | 0.3         | /                |
|                        | Min. water/gas flow ratio (v/v)                                                               | 0.76      | 0.76      | 0.75        | 0.92             |
|                        | Carbonate particle size (μm)                                                                  | 250 – 500 | 250 – 500 | 4           | 10               |
| ency                   | Max sequestration efficiency (%)                                                              | 100       | 100       | 40          | /                |
|                        | CO <sub>2</sub> dissolution efficiency (%)                                                    | 57        | 33        | 63          | 93               |
|                        | CO <sub>2</sub> uptake efficiency (%)                                                         | 57        | 33        | 25          | /                |
| <u>:</u>               | CaCO <sub>3</sub> dissolution efficiency (%)                                                  | 1         | 22        | 48          | 59               |
| eff                    | CO <sub>2</sub> sequestration efficiency (%)                                                  | 0.6       | 6         | 8           | /                |
| Process efficiency     | pH before/after degassing                                                                     | 6.4/8.1   | 6.6/8.2   | 6.7/8.5     | 6.6*/8.0**       |
|                        | Water efficiency factor                                                                       | 4.6       | 3.2       | 0.4         | /                |
|                        | Volume of water used per tonnes of CO <sub>2</sub> captured (10 <sup>3</sup> m <sup>3</sup> ) | 2000      | 150       | 17          | 2                |

#### 3.1. One-step fixed-bed reactor

The first AWL reactor design comprised a one-step fixed-bed reactor (Fig. 3a), of which the theoretical concept was first presented in Rau and Caldeira (1999), and experimental results from a bench-top version were reported in Rau (2011). This reactor contains a porous bed of limestone particles, sprayed with water until they are submerged. The CO<sub>2</sub>-rich gas enters through one or more inlets located at the bottom or lower half of the reactor

(Fig. 3a). Subsequently, the gas stream passes over and through the wetted, porous bed of limestone particles, which then allows the CO<sub>2</sub> in the gas phase to hydrate in the pore fluid. The flue gas (partially) depleted in CO<sub>2</sub> leaves the reactor from the top and is discharged to the atmosphere.

As indicated by the analysis above, the CO<sub>2</sub> uptake from the gas is critically dependent on the water to gas flow ratio  $(Q_w/Q_p)$  - see Eq (22). This was confirmed by laboratory experiments with a bench-top version of the onestep fixed-bed reactor (Rau, 2011). At a low  $Q_w/Q_g$  of below 1, the CO<sub>2</sub> uptake efficiency remained below ~30%, but could be increased up to 97% by increasing the  $Q_w/Q_g$  to >8. Chou et al. 2015 examined a similar lab-scale one-step reactor, and achieved a CO<sub>2</sub> uptake efficiency of ~57 % using a  $Q_w/Q_g$  of 3.5 (Table 2). The dissolution 350 of CO2 in the process water generates a low-pH carbonic acid solution which then can react with the carbonates to form Ca<sup>2+</sup> and HCO<sub>3</sub>. The removal of CO<sub>2</sub> from the flue gas alone however does not imply that the reaction with limestone is completed. Rau (2011) found that the majority of the hydrated CO<sub>2</sub> did not react with the CaCO<sub>3</sub> particles, and would be outgassed again to the atmosphere upon release. This was confirmed by a lab-scale onestep reactor investigated by Chou et al. (2015), which showed a very low CaCO3 dissolution efficiency of only 355 ~1 % (Table 2). Consequently, the overall CO<sub>2</sub> sequestration efficiency of a one-step reactor remains low due to a lack of CaCO<sub>3</sub> dissolution. A large fraction of the dissolved CO<sub>2</sub> remains unbuffered by the increase in A<sub>T</sub>. This unbuffered CO<sub>2</sub> will escape if the solution is exposed to the atmosphere during the re-equilibration step (Rau, 2011; Chou et al., 2015). With such a low CaCO<sub>3</sub> dissolution efficiency, the reactor configuration of Chou et al. (2015) requires an excessive ~2 million m<sup>3</sup> of water to sequester 1 tonne of CO<sub>2</sub> (Table 2). Possibilities to improve 360 the CaCO<sub>3</sub> dissolution efficiency are to increase the reaction time or to decrease the limestone particle size as to increase the reactive surface area and dissolution rate (Rau, 2011).

#### 3.2. Two-step reactor




A fundamental problem of a one-step reactor is that the time scale of CO<sub>2</sub> dissolution is much smaller than that of CaCO<sub>3</sub> dissolution, thus leading to a low CaCO<sub>3</sub> dissolution efficiency. To accommodate this, a two-step reactor design was tested to improve the CaCO<sub>3</sub> dissolution efficiency (Chou et al., 2015). In this, the dissolution of CO<sub>2</sub> in the process water and the CaCO<sub>3</sub> dissolution occur in two separated reactors placed in series (Fig. 3b). In the first step, the CO2-rich gas stream is brought into contact with the inlet process water in a gas-liquid reactor and after the pH of the process water is stabilized, the acid solution was fed into a liquid-solid reactor filled with limestone powder (>95 wt.% CaCO<sub>3</sub>) with a particle size of 250 – 500 μm (Chou et al., 2015). Under identical operation conditions, the CaCO<sub>3</sub> dissolution efficiency could be increased from 1% in the one-step process to 22% in the two-step process (Chou et al., 2015). This reduced the required amount of water needed to sequester 1 tonne of  $CO_2$  to ~150.000 m<sup>3</sup> (Chou et al., 2015).

Figure 3: Conceptual reactor design of four AWL reactors. (a) One-step reactor, (b) Two-step reactor, (c) Slurry reactor, (d) Buffered AWL reactor. SL = slaked lime pipe, DR = dissolution reactor, BR = buffering reactor.

#### 3.3. Slurry reactor






The next improvement in reactor design was achieved by using a suspension of fine CaCO<sub>3</sub> instead of a reactor with large CaCO<sub>3</sub> grains (Fig. 3c). This reactor design was implemented in an AWL demonstration plant at a coal-fired power plant in Wilhelmshaven (Germany) that could process up to 200 m<sup>3</sup> h<sup>-1</sup> of flue gas (Kirchner et al., 2020b). The AWL reactor consisted of a five columns (1.95 m high; 0.32 m diameter) packed with plastic packing rings to increase the surface area within the reactor to enhance the dissolution of CO<sub>2</sub> into the water as well as the subsequent CaCO<sub>3</sub> dissolution. A limestone suspension of approximately 0.5% (w/w) was sprayed into the head space of each column. The desulfurized flue gas from the coal-fired power plant entered the columns from the bottom side. The flue gas was channeled through all five columns sequentially to achieve maximal removal of CO<sub>2</sub>. The flue gas leaving the last column was fed back into the chimney of the power plant. These improvements resulted in a CO<sub>2</sub> uptake efficiency between 15 and 55% during the operation of this AWL demonstration plant with the uptake efficiency being inversely proportional to the  $Q_g$ . For a  $Q_w/Q_g$  of 0.3, a CO<sub>2</sub> uptake efficiency of 25% was achieved (Table 2; Kirchner et al., 2020b). At this  $Q_w/Q_g$ , the  $W_{eff}$  is smaller than 1 and the  $Q_w$  limits the maximum achievable CO<sub>2</sub> sequestration efficiency ( $\eta_{seq}^{max} = 40\%$ ). The CO<sub>2</sub> uptake efficiency can be further improved by increasing the  $Q_w/Q_g$ , by increasing the number of reactor columns or by recirculating the gas stream. Note however that all these factors lead to a larger (and hence more costly) reactor setup.

The CaCO<sub>3</sub> dissolution, step (ii), was improved by using a limestone suspension with micronized CaCO<sub>3</sub> particles ( $\sim$ 4 µm) and by improving mixing and turbulence within the reactor by implementation of the plastic packing rings (Kirchner et al., 2020b). This resulted in an A<sub>T</sub> increase from 2 mM in the input stream to 5.6 mM in the effluent water and a CaCO<sub>3</sub> dissolution efficiency of 48% (Table 2; Kirchner et al., 2020b). This then led to a substantially reduced water consumption (17.000 m<sup>3</sup> per tonnes of CO<sub>2</sub> sequestered) compared to the one-step and two-step reactors (Table 2; Kirchner et al., 2020b). When the process was performed in a closed-loop with recirculation of the process water, an A<sub>T</sub> of >10 mM was achieved. This indicated that the contact time between

the limestone suspension and the flue gas was too short in the one-pass setup. Additional columns, elongation of the existing ones, and higher limestone concentrations could be considered for optimization of the reactor design (Kirchner et al., 2020b). The water stream leaving the columns was fed into a sedimentation tank to separate the remaining limestone particles from the process water. The particle-poor overflow water was then fed into the wastewater treatment system of the powerplant (Kirchner et al., 2020b).

#### 3.4. Buffered accelerated weathering of limestone reactor

The feasibility of unbuffered AWL reactors is hindered by the large water requirements ( $10^3 - 10^5$  m³ water per tonnes of CO<sub>2</sub> sequestered) in current reactor designs (Rau and Caldeira, 1999; Rau, 2011; Caserini et al., 2021). This large water requirement is a direct consequence of the low CaCO<sub>3</sub> dissolution efficiency  $\varepsilon_{CaCO_3}$  (as illustrated by Eq. 24-25). To increase the CaCO<sub>3</sub> dissolution efficiency, longer reaction times and thus larger reactors are required, which then also increases capital investment (Rau, 2011; Kirchner et al., 2020b). A second issue of unbuffered AWL reactors, is the outgassing effect  $\varepsilon_{outgas}$ . If the effluent solution is exposed to the atmosphere, excess CO<sub>2</sub> will be degassed until the effluent is in equilibrium with the pCO<sub>2</sub> of the ambient atmosphere. One option would be to avoid this contact with the atmosphere. If the effluent would be directly discharged into the deep sea, the CO<sub>2</sub> storage potential is higher as it avoids extensive degassing. However, this would also lead to acidification of the deeper ocean and associated environmental impacts (Caserini et al., 2021).

To overcome the issues of low CaCO<sub>3</sub> dissolution efficiency, high water requirements and inefficient CO<sub>2</sub> sequestration of unbuffered AWL, the concept of "buffered AWL" has been proposed (Caserini et al., 2021). Buffered AWL reactors have not been physically built or tested, and still reside within the conceptual phase. Buffered AWL consists of four distinct sections: a mixer, a dissolution reactor (DR), slaked lime pipe (SL) and a buffering reactor (BR) (Fig. 3d). The main difference between AWL is the buffering of the unreacted CO<sub>2</sub> by Ca(OH)<sub>2</sub>. In the mixer, CO<sub>2</sub> from the gas stream is mixed with seawater and CaCO<sub>3</sub> particles to form a homogeneous slurry. The CO<sub>2</sub> gas stream enters the mixer from the bottom and is hydrated through a bubble-type absorption column or a packed bed absorption column. A bubble type absorption column would be preferred as the absorption can be 3-10 times faster than in a packed bed column, which reduces the reactor size significantly (Teir et al., 2014; Xing et al., 2022; Zhang et al., 2023). The CO<sub>2</sub>-depleted gas is released at the top of the mixer. Seawater is fed to the mixer from the upper part. This theoretical example assumes a dissolution of 1000 kg of CO<sub>2</sub> in 2000 m<sup>3</sup> process water, at which point the process water is in equilibrium with the flue gas (pCO<sub>2</sub>  $\cong$  0.28 bar) (Caserini et al., 2021).

CaCO<sub>3</sub> particles, with a suitably small diameter ( $<50 \mu m$ ) so that they remain in suspension, are uniformly mixed with the main water stream at the bottom of the mixer before entering into the dissolution reactor (DR). The dissolution rate of the CaCO<sub>3</sub> particles is determined by the size of the CaCO<sub>3</sub> particles, residence time and pressure in the dissolution reactor (Caserini et al., 2021). The primary objective of the DR is to maximize the amount of dissolved CaCO<sub>3</sub> per tonne of dissolved CO<sub>2</sub> in solution (Caserini et al., 2021). The DR consists of a piping system in which CaCO<sub>3</sub> is dissolved into a fully ionic solution during transport to the coastal ocean. The DR can be located on- or offshore. If the DR is constructed offshore, between the coasts and the deeper ocean, the solution flowing down the DR encounters increasing the hydrostatic pressure which improves the dissolution of CaCO<sub>3</sub> (Dong et al., 2018; Caserini et al., 2021). The CaCO<sub>3</sub> dissolution efficiency (step (ii)) of the theoretical example proposed was 59% (Table 2). The solution leaving the DR will be acidic as CO<sub>2</sub> needs to be present in stoichiometric excess to allow full dissolution of the CaCO<sub>3</sub> particles. Therefore, a final buffering in the buffering

reactor (BR) is needed before discharge to the ocean. This BR is located at the end of the DR. Aqueous calcium hydroxide (Ca(OH)<sub>2</sub>), supplied through the slaked lime pipe, is mixed with the acid solution leaving the DR. The Ca(OH)<sub>2</sub> reacts with the unreacted CO<sub>2</sub> remaining in the solution at the end of the DR.

The buffering of the unreacted  $CO_2$  by  $Ca(OH)_2$  allows to release an ionic solution at the same pH as the seawater and thereby avoiding acidification. The buffering also avoids degassing of the unreacted  $CO_2$  and increases the long-term storage efficiency of the process compared to traditional AWL (Caserini et al., 2021; Chou et al., 2015; Rau, 2011). The use of a tubular reactor in the buffered AWL process also allows for long residence times, higher pressures and reduces the need for maintenance. High-density poly-ethylene (HDPE) pipelines have a long lifetime and can be used up to 900 m deep. Extending the DR into the deep sea allows for efficient dissolution of  $CaCO_3$  as dissolution is favored at high pressure. This reduces the amount of  $Ca(OH)_2$  that would be needed to compensate for the unreacted  $CO_2$  left in the solution.

The use of Ca(OH)<sub>2</sub> and micronized CaCO<sub>3</sub> particles comes, however, at an energy and CO<sub>2</sub> penalty. This penalty can be minimized by using electric energy from renewable sources for the production of Ca(OH)<sub>2</sub> and the milling of CaCO<sub>3</sub> (Caserini et al., 2021). Furthermore, Ca(OH)<sub>2</sub> can potentially be made from steel slags at low temperatures lowering the CO<sub>2</sub> emissions by at least 65% (Castaño et al., 2021). The estimated cost for capturing and storing CO<sub>2</sub> using buffered AWL is comparable with estimates for large-scale geological carbon capture and storage projects (De Marco et al., 2023).

#### 4. AWL feedstocks








The three feedstock components needed for traditional AWL are water,  $CaCO_3$ , and  $CO_2$ , with the addition of  $Ca(OH)_2$  in the case of buffered AWL. The amount of materials needed will depend of the pCO<sub>2</sub> in the flue gas and the efficiency of the reactor (Table 2).

Limestone (containing 92 – 98% CaCO<sub>3</sub> (Rau et al., 2007)) is the primary mineral source of CaCO<sub>3</sub> as it is much more abundant and less expensive than pure CaCO<sub>3</sub> (~4\$ tonne<sup>-1</sup> limestone, ~105\$ tonne<sup>-1</sup> dolomite, ~400\$ tonne<sup>-1</sup> pure CaCO<sub>3</sub>; Calcium Carbonate Prices, News, Monitor, Analysis & Demand, 2024; Caserini et al., 2021). The US production of limestone was about 1.05 x10<sup>9</sup> tonnes in 2023 (Survey, 2023), with Sweden being the largest producer in Europe accounting for a production of 6.3 x10<sup>6</sup> tonnes in 2021 (Mineral statistics, 2024). About 20% of the limestone production and processing results in waste limestone fines with no significant market value (Rau et al., 2007; Langer et al., 2009). These fines could be used as a low-cost source of CaCO<sub>3</sub> for application in AWL and at the same time reduce waste from limestone mining and processing.

Significant volumes of water are needed to dissolve the CO<sub>2</sub> and dilute the resulting bicarbonate in the original reactor designs (10<sup>4</sup> - 10<sup>5</sup> tonnes of water per tonne of CO<sub>2</sub>; Table 2) (Rau et al., 2007; Rau and Caldeira, 1999), although more recent designs have reduced the water demand by a few orders of magnitude (~ 10<sup>3</sup> tonnes of water per tonne of CO<sub>2</sub>; Table 2). The high water demand and the accompanying pumping cost could limit the feasibility of the overall AWL process. Therefore, a low-cost water source such as cooling water from a power plant or other sources of recycled water should be used preferably (Rau and Caldeira, 1999). Due the required quantities of process water, the favored locations for (un)buffered AWL reactors would be coastal regions as seawater is a virtually limitless source and the bicarbonate-containing effluent could be directly dumped and diluted in the ocean after degassing or buffering and removal of potential contaminants (Rau and Caldeira, 1999; Rau et al., 2001). Pumping costs could further be reduced by reusing the large volumes of seawater already pumped and used

as power plant cooling water (Rau et al., 2007; Kirchner et al., 2021). However, the elevated temperature of the seawater during the cooling of the power plants would reduce the CO<sub>2</sub> dissolution into the seawater (Kirchner et al., 2021).

The third resource needed in the AWL process is  $CO_2$ . AWL can use different industrial point sources of  $CO_2$ . However, the  $CO_2$  concentration in the flue gas of different industrial sources can vary substantially from  $\sim 3$  vol% in a natural gas turbine up to 25 vol% in cement plants (De Marco et al., 2023). As increased  $CO_2$  concentrations in the gas stream promotes dissolution of  $CO_2$  in the seawater, industrial sources with high concentrations of  $CO_2$  in the flue gas are preferable (De Marco et al., 2023; Rau and Caldeira, 1999).

Buffered accelerated weathering of limestone uses a fourth feedstock, calcium hydroxide (Ca(OH)<sub>2</sub>) also known as slaked lime. The Ca(OH)<sub>2</sub> is used to buffer the remaining unreacted CO<sub>2</sub> at the end of the reactor to be able to release a solution at the same pH as the seawater (Caserini et al., 2021). Slaked lime is produced through calcination of limestone to form calcium oxide (CaO), which is then granulated and hydrated to from Ca(OH)<sub>2</sub> (Castaño et al., 2021; Simoni et al., 2022). This production process generates about 1 – 1.8 tonnes of CO<sub>2</sub> per tonne of Ca(OH)<sub>2</sub> (Oates, 2008; Simoni et al., 2022). This results in CO<sub>2</sub> penalty for the buffered AWL process. However, if Ca(OH)<sub>2</sub> can be made from alkaline industrial waste, such as steel slag, through a calcination-free pathway, the specific CO<sub>2</sub> intensity can be reduced by as much as 65% (Castaño et al., 2021). This will greatly improve the CO<sub>2</sub> sequestration efficiency of the buffered AWL process.

Due to the high resource requirements especially for process water and CaCO<sub>3</sub>, the (un)buffered AWL plant should preferably be located near the coast and close to limestone deposits and mines. This will reduce the economic and environmental cost of long distance transport of large volumes of water and limestone and thereby increase the overall efficiency of the (un)buffered AWL process (Kirchner et al., 2021; Rau et al., 2007).

#### 5. Environmental concerns








Seawater is the preferrable source of process water for AWL as it requires large volumes of water. The intake of large volumes of seawater could lead to entrainment and impingement of small marine organisms (Liyanaarachchi et al., 2014; Missimer and Maliva, 2018). To avoid additional environmental damage to marine organisms from seawater intake, downstream seawater discharge of cooling water from power plant facilities could be used. This combined water usage has several benefits which include: 1) avoidance of the need to build expensive offshore intake structures, 2) no need for maintenance of the offshore infrastructure, 3) avoid extra potential damage from seawater intake, and 4) minimal need for environmental permitting as primary intake is already permitted (Liyanaarachchi et al., 2014).

During the process of AWL, large amounts of effluent water will be produced that needs to be discharged in rivers or coastal areas. As seawater is a preferred source of process water used in AWL, disposal of the effluent water in the ocean will be the most likely option. Considering the large pool of DIC already present in the ocean and the natural variability of  $A_T$  on diurnal, seasonal, and interannual basis, the discharge of AWL effluent water can be expected to only have minor effect on  $A_T$  and DIC concentrations (Rau et al., 2007; Kirchner et al., 2020a). Nevertheless, changes in the balance between  $A_T$  and DIC induced by AWL discharge can affect pH and the calcite and aragonite saturation state ( $\Omega_{calc}/\Omega_{aragonite}$ ) (Chou et al., 2015; Kirchner et al., 2020a), which in turn can impact the calcification rate of several major groups of marine calcifiers such as coccolithophores, foraminifera and corals, in a similar fashion as ongoing ocean acidification (Kleypas et al., 1999; Ries et al., 2009). However,

the pH in coastal ecosystem can vary strongly in space and time. In vegetated areas, photosynthesis, and respiration cause significant change in the environmental pH on a diurnal time scale (0.2-0.7 pH units; Hendriks et al., 2014; Rivest and Gouhier, 2015; James et al., 2020), with the largest pH fluctuations found in sheltered areas with low hydrodynamics (James et al., 2020). Therefore, it is important to consider the local ecosystem and hydrodynamic regime to estimate the effect the discharge water will have on the local environment. The effluent pH from the reactors analyzed here are in the range 6.4-8.5 (Table 2). If the effluent with a pH of 6.5 were discharged directly into the ocean, the expect acidification impact would be significant. To limit environmental effects, the effluent could be diluted with seawater before discharge. A 10-fold dilution would be sufficient to bring an effluent pH of 6.5 back to within the tolerable range of < 0.2 pH units change from background levels (Chou et al., 2015). Discharge in a place with strong currents would be favorable to achieve rapid advection and mixing between the discharge water and the receiving seawater (Chou et al., 2015). Inversely, if the effluent water is allowed to equilibrated with the atmosphere before discharge, or buffered with Ca(OH)<sub>2</sub>, the increased A<sub>T</sub> and pH would help counter ocean acidification and its effect on marine biota (Rau and Caldeira, 1999; Rau et al., 2007; Chou et al., 2015; Albright et al., 2016; Kirchner et al., 2020a; Sánchez et al., 2024).

Another environmental concern is the potential release of impurities from the limestone or flue gas. In particular if flue gas from coal-fired power plants would be used, as this is known to contain SO<sub>x</sub>, NO<sub>x</sub>, and trace elements (Rau et al., 2007; Kirchner et al., 2020a, b). The dissolution of SO<sub>x</sub> and NO<sub>x</sub> can lead to the formation of strong acids such as H<sub>2</sub>SO<sub>4</sub>, HNO<sub>3</sub>, and HNO<sub>2</sub>. These dissolution products can lead to eutrophication and reduced biodiversity, if discharged directly in the aquatic environment. Existing flue gas desulfurization facilities already in use at most power plants can effectively remove most of the SO<sub>x</sub> contained in the flue gas. The solubility of NO<sub>x</sub> is fairly limited and most will leave with the CO<sub>2</sub>-depleted gas stream leaving the AWL reactor. The effluent stream of an AWL pilot plant utilizing desulfurized flue gas contains SO<sub>4</sub><sup>2-</sup> and N-species in concentrations below the marine background level (Kirchner et al., 2020b). Trace elements such as Ba, Co, Ni, and Zn could be released from the flue gas or from the dissolution of CaCO<sub>3</sub>, while increased concentrations of Mn and Co were found in the effluent stream of the AWL plant in Wilhelmshaven (Germany). However, the final concentrations were not expected to be of environmental concern and well below the environmental guidelines (Kirchner et al., 2020b). The potential negative effects from trace elements and other pollutants can be further mitigated by using of relatively clean waste gas streams (such as from the combustion of natural gas or calcination of CaCO<sub>3</sub>) in (un)buffered AWL applications.

The disposal of large volumes of process water in the surface water of the coastal zone can locally increase pH and mitigate the adverse effect of ocean acidification on calcifying phytoplankton. However, this implies a reduction of the efficiency of the CO<sub>2</sub> sequestration via AWL, as part of the produced A<sub>T</sub> will be consumed and lead to CO<sub>2</sub> degassing (Lehmann and Bach, 2025). Additionally, mixing of this A<sub>T</sub> enriched coastal water within the coastal sediment through porewater flushing or diffusion could potentially inhibit natural CaCO<sub>3</sub> dissolution (Lunstrum and Berelson, 2022; Bach, 2024). If this would occur, the efficiency of the (un)buffered AWL process would be reduced as the CO<sub>2</sub> sequestration by AWL would be partially compensated by a loss of natural CO<sub>2</sub> sequestration. However, this is less likely to occur with (un)buffered AWL than with mineral-based OAE where alkaline minerals are directly added to the coastal sediment and A<sub>T</sub> can build-up in the porewater (Hartmann et al., 2023).

#### **Summary and conclusions**








Accelerated weathering of limestone (AWL) is a CO<sub>2</sub> emission mitigation technology that aims to artificially increase the weathering rate of CaCO<sub>3</sub> (Rau and Caldeira, 1999). The AWL process consist of four main steps: (i) The CO<sub>2</sub> uptake step, (ii) the CaCO<sub>3</sub> dissolution step, (iii) the alkalinization step (for buffered AWL), and (iv) the re-equilibration step.

Since the first AWL reactor design proposed by Rau and Caldeira in 1999 (Rau and Caldeira, 1999), laboratory experiments and pilot scale operations have optimized the CO<sub>2</sub> uptake efficiency and reduced resource consumption. Nevertheless, large quantities of water are still needed for the dissolution of CaCO<sub>3</sub>, while degassing of CO<sub>2</sub> after contact of the effluent with the atmosphere limits the CO<sub>2</sub> sequestration efficiency. The concept of buffered AWL, as proposed by Caserini et al. (2021), reduces the water requirements and increases the CO<sub>2</sub> sequestration efficiency by adding an extra Ca(OH)<sub>2</sub> buffering step. This additional step however comes at a CO<sub>2</sub> penalty, as conventional production of Ca(OH)<sub>2</sub> emits CO<sub>2</sub>.

Improved design of reactors and generation of feedstock can further optimize the CO<sub>2</sub> sequestration efficiencies. The tubular reactor design used in buffered AWL reduces the required reactor size significantly compared to traditional unbuffered AWL reactors. The use of a tubular reactor furthermore allows for long residence times and higher pressures which stimulates CaCO<sub>3</sub> dissolution (Caserini et al., 2021). Furthermore, using renewable energy and starting from waste limestone fines for the milling of CaCO<sub>3</sub> particles and producing Ca(OH)<sub>2</sub> from alkaline industrial waste via calcination-free processes can avoid the CO<sub>2</sub> penalty of buffered AWL (Caserini et al., 2021; Castaño et al., 2021). The pumping of the large quantities of process water needed in (un)buffered AWL requires a significant amount of energy. Therefore, optimization of the water usage is needed and could be achieved by increasing the pressure of the incoming gas stream or increasing the fraction of CO<sub>2</sub> in the gas stream. Reusing the cooling water from nearby power plant could further reduce costs and environmental damage associated with large water intake. Further optimization of the dissolution kinetics of the micronized CaCO<sub>3</sub> particles could reduce the amount of Ca(OH)<sub>2</sub> needed in the buffering and thereby reducing the energy and CO<sub>2</sub> penalty from the Ca(OH)<sub>2</sub> production.

The effects of disposing large amounts of effluent with increased  $A_T$ , altered pH, and trace elements to the marine environment are currently poorly constrained. Existing research on ocean acidification and ecotoxicological studies on trace element toxicity can provide information of ecosystem impacts of AWL water discharge. However, because of the limited number of operational pilot plants, little is known about the actual conditions that can be expected for AWL water discharge. If AWL is to be implemented as a  $CO_2$  emission mitigation technology on a large scale in the next decade, more pilot plants should be constructed sooner rather than later.

## Code availability

The code used for analysis in this study is available upon request. Interested parties may contact the corresponding author.

#### **Author contribution**

FJRM conceptualized the idea for the review. TH was responsible for data collection, analysis and visualisation and SJVDV supervised the study. TH and FJRM worked out the model formulation. TH has written the manuscript with substantial contribution from all authors.

## 595 Competing interests

The authors declare that they have no conflict of interest.

## Financial support


This research was supported by the Research foundation Flanders (FWO-SBO project S000619N) and the VLAIO Blauwe cluster project "Blue Alkalinity" (HBC.2023.0496). Additional financial support was provided by Team for the Planet (https://team-planet.com), a citizen community initiative dedicated to tackling of the climate challenge (funding through a collaboration agreement with its Carbon Time subsidiary).

## Acknowledgements

The authors thank Gunter Flipkens for academic discussions.

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
