# Peer review of "Reviews and syntheses: Potential and limitations of oceanic carbon dioxide storage via reactor-based accelerated weathering of limestone"

_EGUsphere, 2025_

## Author Comment (AC1)

**Reply to referee 1**

We would like to thank the reviewer for their constructive and positive feedback on our manuscript. Their recommendations have significantly improved the structure and content of the text. Below we provide a response to all their comments and suggestions, and indicate how we have altered the manuscript in response; our responses are in blue, altered text is in shaded in grey.

1. Line 35 Use of CCS as the overarching term for point-source $CO_2$ mitigation is inappropriate because CCS has come to mean a very specific form of that mitigation https://en.wikipedia.org/wiki/Carbon_capture_and_storage

   CCS is replaced by $CO_2$ emission mitigation throughout the manuscript.

2. Line 79-81 "The concept of AWL was first proposed by Rau and Caldeira more than two decades ago (Rau and Caldeira, 1999). It provides a geochemistry-based method for CCS in which the dissolution of carbonate minerals is artificially enhanced (Rau and Caldeira, 1999)."

   You mean -

   The concept of AWL was first proposed more than two decades ago by Rau and Caldeira (1999). It provides a geochemistry-based method for CO2 emissions mitigation in which the aqueous reaction of carbonate minerals with CO2 is enhanced due to the elevation of CO2 in typical waste combustion gases (Rau and Caldeira, 1999). ?

   The sentences are adjusted according to the recommendations of the reviewer.

3. Line 90 Cite Caserini et al (2021) in initially introducing/describing BAWL.

   The citation is added.

4. Table                                                                                              1

   Row 1 - The initial values here are very uncharacteristic of low latitude, surface SW. Chou et al. et al 2015 are referenced as the source, and the values appear to be taken from their Table 1 (representing offshore and probably deep water samples) although I don't see the specific At and DIC values used by the present authors. In any case, it is clear from Chou et al. et al Table 1 that the starting solutions were not air equilibrated, pCO2>700 uatms, thus DIC is elevated and pH and Omega are depressed. The more realistic starting conditions are listed in Chou et al. eta Table 2 where pH>8 and esp Omega(c) >4.5. The choice of starting conditions will have a very significant effect on the modeling outcomes of the present study, so I ask the authors to carefully justify their initial choice of values here.

   Row 2 The amount of DIC rise in equilibrium with 0.15atm CO2 will very much depend on the chemistry of the starting solution that I question above.

Row 3 Ditto. Why does Omega(c) only rise to 0.203? In a perfect world under full CO2 and CaCO3 equilibrium OmegaC = 1. Granted, the kinetics for reaching this equilibrium are too slow to be reached in a practical application, but why is CaCO3 dissolution stopped at OmegaC=0.203 when the solution is still significantly carbonate undersaturated? The ratio of DeltaDICseq/DeltaDICcarb = 0.83/0.19 = 4.4. Shouldn't this be closer to 1? Or is there a huge amount of excess, unreacted CO2aq in solution?

Rows 4 and 5 Values are highly dependent on the accuracy of the preceding conditions/modeling.

The reviewer is correct in his assertion that the outcomes of our thermodynamic modelling are dependent on the initial conditions. Therefore, we did not use hypothetical 'ideal' starting conditions – as these 'ideal' starting conditions are also location dependent, and would thus vary for each potential AWL reactor. Instead, we used data from published pilot studies. The aim of Table 1 was thus to give an example of the different states for a representative real life (bench-top) reactor setup. All the values calculated in Table 1 are based on the measured inlet and outlet $A_T$ and DIC from the two-step bench-top reactor from Chou et al. (2015).

Note that the initial solution values from Table 2 from Chou et al. (2015) cannot be used for calculations as there are no measured values at the outlet for these starting conditions.

To avoid this confusion, we clarified the purpose of Table 1 upfront at Line 111-117:

*"Table 1 shows the values for $pCO_2$, $A_T$, DIC, pH and $\Omega_{calc}$ in each of the four states for a representative case study, which is based on data reported from a two-step bench-top reactor consisting of a separate gas-liquid and liquid-solid reactor (Chou et al., 2015, further discussed below). The $CO_2$ concentration of the gas stream was 15%, while the $pCO_2$ of the atmosphere is fixed at 420 ppm. The $A_T$ and DIC values at the inlet and outlet of the reactor are based on measured values (Table 1 in Chou et al., 2015). The remaining variables are calculated using the CRAN:AquaEnv package for the thermodynamic equilibria of acid-base systems in seawater (Hofmann et al., 2010)."*

And expanded the caption of Table 1:

*"Theoretical values for alkalinity ($A_T$), dissolved inorganic carbon (DIC), pH and calcite saturation state ($\Omega_{calc}$) in the four consecutive states of the example AWL reactor: (1) the process water that is used as intake (the process water was collected from an offshore station near the Hoping power plant and the inlet and outlet of the cooling water drainage of the Hoping power plant (Chou et al., 2015)) , (2) the process water with elevated DIC after $CO_2$ uptake, (3) the process water enriched in $A_T$ and DIC after $CaCO_3$ dissolution, (4a) the unbuffered or (4b) buffered process water upon discharge. $\Delta DICseq$ is the DIC that is added to the process water due to dissolution from the gas stream and $\Delta DICcarb$ is the DIC added through the dissolution of $CaCO_3$. The $pCO_2$, $A_T$ and DIC values (indicated by #) are based on values measured in a two-step AWL bench-top reactor (Chou et al., 2015). The values of $A_T$, DIC, pH, and $\Omega_{calc}$ (indicated with *) are calculated using CRAN:AquaEnv (Hofmann et al., 2010) for seawater at a temperature of 15 °C and salinity of 35."*

The reason Omega only rises to 0.203 at the reactor outlet, is because the dissolution reaction was too slow to completely buffer the saturation state drop. The omega is calculated for the

values at the outlet given by Chou et al. and are before re-equilibration. We clarified this observation at Line 147 – 149:

> *"Note that the effluent at state 3 of the example two-step reactor was not in equilibrium with $CaCO_3$ dissolution ($\Omega_{calc} < 1$, Table 1), which indicates that the effectiveness of $CaCO_3$ dissolution in the reactor design of Chou et al. (2015) could still be improved."*

5. Equ 1 Only valid at low pH (<7). The stoichiometry changes as pH rises so as to accommodate the spontaneous formation of (alkalinity hog) CO3-- ; ACO2 + BH2O + CaCO3 ---> Ca++ CHCO3- + DCO3-- + ….. such that the total moles carbon added is A+1=C+D (<=2) and A<=1 (see eq 1 here https://bg.copernicus.org/articles/20/27/2023/)

This is correct, in a sense that the carbonate system re-equilibrates after the reaction, which will set the eventual stoichiometry of the overall reaction. However, we think this leads to a confusing way of writing the equations, as A, B, C, … are dependent on conditions and obscure the fact that the dissolution of calcium carbonate always creates two alkalinity. We have chosen to explain the re-equilibration step (see Line 133 of the original manuscript), and if one would combine Eq. (1) and Eq. (2) for a given pH, you would get the equation the reviewer refers to. We prefer to keep our approach, as we think this makes the overall process clearer for non-specialist readers.

6. Line 143-4 "However, one can easily show that equilibration followed by mixing, provides the same CO2 transfer as mixing followed by equilibration." This assumes that discharing a supersaturated CO2 solution into seawater will in fact equilibrate with air (on human-relevant timescales). That is unlikely to happen due the slow kinetics of air/sea gas exchange coupled with vertical SW mixing that will remove some of the supersaturated solution out of contact with air prior to equilibration. Gorey details here: https://www.nature.com/articles/s41558-024-02179-9

Bottom line: Assuming air equilibration underestimates C storage because some excess CO2aq added in unbuffered AWL will not have a chance to degas to air.

Full equilibration with the atmospheric $pCO_2$ will indeed be prevented when surface residence times of the discharged process water is shorter than the air-sea equilibration timescale. So, when the process water is discharged below a strong stratification layer or in locations where the discharged water quickly reaches the deeper oceans, assuming full equilibration would underestimate the $CO_2$ storage potential (He & Tyka, 2023; Jones et al , 2014). However, as most AWL plants, like the AWL pilot plant in Wilhelmshaven (Germany; Kirchner et al., 2020), will be located near the coastal ocean with shallow mixed layers with relatively efficient air-sea $CO_2$ exchange, equilibration will take place on timescales of months up to a year (Jones et al, 2014; Geerts et al, 2025).

We added the caveat about air-sea $CO_2$ exchange variation and the possibility at non-equilibrium degassing at Line 117-125:

> *"We assume full re-equilibration with the atmosphere (unbuffered AWL) or full buffering with $Ca(OH)_2$ upon discharge into the sea (buffered AWL). In the well-mixed coastal zone, air-sea $CO_2$ exchange takes place on a time-scale of several weeks up to a year (Jones et al., 2014; He and Tyka, 2023; Geerts et al., 2025). However, when the surface residence time of the discharged process water is shorter than the air-sea $CO_2$ equilibration timescale, some of the*

*dissolved $CO_2$ which is unbuffered by an $A_T$ increase will move to deeper layers and full re-equilibration will not be reached (Jones et al., 2014; He and Tyka, 2023). When the process water is discharged below a strong stratification layer or directly in the deeper ocean full re-equilibration will also be prevented (Jones et al., 2014; He and Tyka, 2023). Therefore, assuming full re-equilibration represents a conservative lower bound of the $CO_2$ sequestration during AWL."*

7. Fig 2 Should be modified depending on the (new) outcomes listed in Table 1.

   See our response to comment 4

8. Line 178-80. "In a similar fashion, the final alkalinity value is the result of alkalinity addition during carbonate dissolution and possibly some extra addition during lime buffering"

   Unclear. If you are adding lime you are adding alkalinity, no "possibly" about it. Or are you saying that adding lime is a possibility? In this region of the text the discussion seems to move from AWL with an option to lime to one where liming is now assumed/required. Please be clear from the start about how you are treating AWL +/-liming.

This phrasing is indeed a bit confusing. We adjusted the sentence to make it clear that we mean that the final alkalinity is the result from carbonate dissolution, with extra alkalinity added by lime buffering in BAWL at Line 196-198

"In a similar fashion, the final alkalinity value is the result of alkalinity addition during carbonate dissolution and the alkalinity that is added during buffering with $Ca(OH)_2$ in the case of BAWL."

9. Line193-4 Full air equilibration after discharge is unlikely (https://www.nature.com/articles/s41558-024-02179-9)

See our response to comment 6.

10. Equ 8 Missing an operator between the 2$^{nd}$ the 3$^{rd}$ right hand terms?

A multiplication sign is added between the two terms for clarification.

11. Line 196-204. Assumes full air/sea CO2 equilibration, unlikely (https://www.nature.com/articles/s41558-024-02179-9)

   See response to comment 6.

12. Line 219-225 Revise depending on outcomes in (revised) Table 1?

See our response to comment 4

13. Line 297-and after Flows and efficiencies are calculated from data in Table 2 with the implication that these values will be characteristic of AWL at scale, yet what is the evidence that the data in Table 2 represent optimized systems?

The idea of Table 2 is to calculate efficiency values for different existing/conceptual reactor designs. Since we are reviewing the existing literature, it is not our goal to represent optimized systems at scale. The operational stage of each specific example reactor is specified in row 1 of the operational conditions. To prevent misunderstanding, we clarified that these are values for prototype/conceptual reactors and that the efficiencies are calculated based on the inlet an outlet $A_T$ and DIC, and the given water/gas flow rate at Line 321 – 325:

> *"The operational conditions and process efficiencies of these reactor designs are summarized in Table 2. The presented operational conditions are given for specific example reactor setups (bench-top (Chou et al., 2015) or pilot plant (Kirchner et al., 2020b)) or conceptual designs (Caserini et al., 2021) and the process efficiencies are calculated based on published data for a specific operational condition. Changes in reactor design or operational conditions will change these calculated efficiencies."*

14. Line 258 You mean 150,000 m^3, yet eq 20 is in units of tonnes/tonne and the assumes that 1L SW = 1kg?

Correct, the exponent should have been 3 (150 000 $m^3$) instead of 2 (150 000 $m^2$). This has been changed.

The units for eq. 20 at $m^3_{seawater}$/tonne of $CO_2$ , as is stated in the text: *'The volume of process water ($m^3$) that is used to capture one tonne of $CO_2$'*. We did notice a typo, $10^6$ has to be $10^{-6}$.

$\Delta DIC_{seq}$ is expressed in mol per unit of volume: $mM = \dfrac{10^{-3} mol}{L} = \dfrac{10^{-3} mol}{dm^3} = \dfrac{10^{-3} mol}{10^{-3} m^3} = \dfrac{mol}{m^3}$

The units for Eq. 20 are: $\dfrac{1}{mol_{CO_2}/m^3_{seawater}} \dfrac{10^{-6}}{g_{CO_2}/mol_{CO_2}} = \dfrac{10^{-6}}{g_{CO_2}/m^3_{seawater}} = 10^{-6} \dfrac{m^3_{seawater}}{g_{CO_2}} = \dfrac{m^3_{seawater}}{10^6 g_{CO_2}} = \dfrac{m^3_{seawater}}{tonnes_{CO_2}}$

15. Line 293-6 What is the evidence that the efficiencies stated are representative of optimized systems?

See response to comments 4 and 13.

16. Line 301-2 You likely mean Rau (2011) rather than Caldeira and Rau (2000)? The former pub offers numerous results/data for a one step reactor. Compare/contrast with Chou et al. et al 2015 and you subsequent calcs?

The citation should indeed be Rau (2011). Results from Rau (2011) and Chou (2015) are compared in section 3.1 but comparing specific calculations is not possible as specific values for DIC, $A_T$, water/gas flow rate are not specified in Rau (2011).

17. Line 324-5 This does not jibe with Rau (2011) which states "Comparing resulting DIC and alkalinity to that of the original solutions and to ambient seawater demonstrates that 61-85% of the carbon originally added to the seawater remained in solution (Figure 2c), with

little change in alkalinity and with no visual evidence of carbonate precipitation after aeration."

*The section in Rau (2011) that the reviewer refers to discusses the modified reactor in which seawater in equilibrium with the $CO_2$/air mixture was allowed to reside in the reactor for 1-2 weeks. This in essence becomes a two-step reactor with long residence time in the second reactor and is thus not applicable on the one-step reactor.*

18. Line 327-8 "Consequently, the overall CO2 sequestration efficiency of a one-step reactor remains low due the lack of conversion from hydrated CO2 to HCO3-." Hydrated CO2 is HCO3- + H2CO3. What is apparently meant here is lack of conversion of hydrated CO2 balanced by Ca++ rather than by H+? Or do you mean lack of conversion of CO2 to carbonic acid? Anyway, how does this square with the 61-85% of the initially captured C shown to be air stable by Rau (2011)?

*For the comparison with Rau (2011) see response to comment 17.*

*We agree with the reviewer that this formulation does not clearly represent the limiting step of $CaCO_3$ dissolution and the production of alkalinity. We have changed the sentence to better convey that we mean the buffering of the dissolved $CO_2$ by the increase in $A_T$ at Line 348 – 350:*

> *"Consequently, the overall $CO_2$ sequestration efficiency of a one-step reactor remains low due lack of $CaCO_3$ dissolution. A large fraction of the dissolved $CO_2$ remains unbuffered by the increase in $A_T$."*

19. Line 417-19 If the now alkalized and carbonated SW is discharged at the same pH as ambient SW the pCO2 must be higher than ambient? Don't you need to discharge at higher pH to avoid this? And wouldn't higher discharge pH beneficially help counter ongoing ocean acidification?

*The reviewer raises a good point. Based on the alkalinity and DIC values at the end of the buffering reactor (BR) from Table 1 in Caserini et al. (2021), and given a seawater temperature of 10 °C, the pH and $fCO_2$ of the process water can be modelled using CRAN:AquEnv.*

*Under these conditions, the pH is 8 (as in Table 1 in Caserini et al. (2021)) and the $fCO_2$ is 0.00483 atm or 4830 µatm. Thus, under conditions presented by Caserini et al. (2021) the $fCO_2$ is indeed higher than ambient if the pH is at the same level as the surrounding seawater.*

20. Line 435-332 Check out Langer et al for further discussion of limestone sources (in the US): https://www.researchgate.net/publication/283868780_Accelerated_weathering_of_limestone_for_CO2_mitigation_Opportunities_for_the_stone_and_cement_industries

*We thank the reviewer for this resource, we have now included it in our reference list.*

21. Line 443-6 Here and elsewhere "high water demand" is implied to be an AWL showstopper, yet the global supply of seawater seems rather limitless. What is apparently meant here is that the pumping costs of seawater can become prohibitive, yet so far no discussion of exactly what these costs are, especially relative to the (high) cost of the industry darling, CCS – capturing, concentrating and storing molecular CO2 underground.

We agree with the reviewer that the supply of seawater should not be seen as a limiting factor, we have adjusted the text accordingly at Lines 464 – 475:

> *"Significant volumes of water are needed to dissolve the $CO_2$ and dilute the resulting bicarbonate in the original reactor designs ($10^4$ - $10^5$ tonnes of water per tonne of $CO_2$; Table 2) (Rau et al., 2007; Rau and Caldeira, 1999), although more recent designs have reduced the water demand by a few orders of magnitude (~ $10^3$ tonnes of water per tonne of $CO_2$; Table 2). The high water demand and the accompanying pumping cost could limit the feasibility of the overall AWL process. Therefore, a low-cost water source such as cooling water from a power plant or other sources of recycled water should be used preferably (Rau and Caldeira, 1999). Due the required quantities of process water, the favored locations for (B)AWL reactors would be coastal regions as seawater is a virtually limitless source and the bicarbonate-containing effluent could be directly dumped and diluted in the ocean after degassing or buffering and removal of potential contaminants (Rau and Caldeira, 1999; Rau et al., 2001). Pumping costs could further be reduced by reusing the large volumes of seawater already pumped and used as power plant cooling water (Rau et al., 2007; Kirchner et al., 2021). However, the elevated temperature of the seawater during the cooling of the power plants would reduce the $CO_2$ dissolution into the seawater (Kirchner et al., 2021)."*

22. Line 447-8 Who has proposed the use of anything but seawater for AWL? The only places AWL will work are near the ocean, eps powerplants that use SW for cooling(?)

The different possible water resources were suggested in Rau and Caldeira (1999) on page 1807 in section 4: Water considerations.

AWL will indeed only be possibly economically feasible near the ocean and indeed especially when seawater used for cooling in power plants can be reused, limiting pumping costs. We have elaborated this paragraph to make it clear that seawater is the only viable option and that pumping costs could further be reduced by reuse of power plant cooling water, as outlined in our response to comment 21.

23. Line 458-9 "The BAWL reactor setup proposed by Caserini et al. consumes 0.4 tons of Ca(OH)2 to store 1 ton of CO2." Or 1/0.4 = 2.5 t CO2/t Ca(OH)2(?) Yet the delta DIC/deltaAlk in the surface ocean is about 0.85. Since 1 mole of Ca(OH)2=2 moles Alk, then the mole CO2 captured and stored per mol Ca(OH)2 should be 2x0.85/1 = 1.7 moles/mole. CO2= 44g/mol, Ca(OH)2= 74g/mol, Thus ,1 tonne of Ca(OH)2 is able to capture and store about 1.7x44/74 = 1 t CO2/tCa(OH)2 in seawater @pCO2= 420 uatms?  Or does 2.5 t/t only apply to deep ocean, high pressures?

This sentence could lead to misunderstanding. We meant that 0.4 tons of Ca(OH)$_2$ was used on top of the 1.31 tonnes of CaCO$_3$ that was fully dissolved, as stated by Caserini et al. (2021). We decided to remove this sentence to avoid confusion.

24. Line 466-8 Seems pretty obvious from the previously published lit. Why even hint at the use of other water sources?

See response to comment 22.

25. Line 499-500 "..the increased alkalinity and pH could potentially limit ocean acidification.." You mean "…the increased alkalinity and pH would help counter ocean acidification and

its effect on marine biota, see for example Albright et al (2016)"
https://www.nature.com/articles/nature17155

The sentence is adjusted according to the comment of the reviewer.

26. Line 514 How about inserting "All of the preceding argue for the use of relatively clean waste gas streams (such as from the combustion of natural gas) in (B)AWL applications." ?

We have added the sentence.

27. Line 515-20 Bach (2024) specifically discusses the application of alkaline solids to marine sediments and the effect of alkalinity generation there. Discharge of dissolved alkalinity into surface waters some distance from sediments and with rapid dilution, as characteristic of (B)AWL, would seem to pose much less risk to benthic/sediment processes.

The discharge of dissolved alkalinity pose less risk than addition of alkaline solids to the sediment. We have highlighted this in this section and further discussed the potential negative feedback in the water column based on the recently published manuscript by Lehman & Bach (2025) at Line 541 - 549:

> *"The disposal of large volumes of process water in the surface water of the coastal zone can locally increase pH and mitigate the adverse effect of ocean acidification on calcifying phytoplankton. However, this implies a reduction of the efficiency of the $CO_2$ sequestration via AWL, as part of the produced $A_T$ will be consumed and lead to $CO_2$ degassing (Lehmann and Bach, 2025). Additionally, mixing of this $A_T$ enriched coastal water within the coastal sediment through porewater flushing or diffusion could potentially inhibit natural carbonate dissolution (Lunstrum and Berelson, 2022; Bach, 2024). If this would occur, the efficiency of the (B)AWL process would be reduced as the CO2 sequestration by AWL would be partially compensated by a loss of natural $CO_2$ sequestration. However, this is less likely to occur with (B)AWL than with mineral-based OAE where alkaline minerals are directly added to the coastal sediment and alkalinity can build-up in the porewater."*

References

He & Tyka, 2023: https://doi.org/10.5194/bg-20-27-2023

Jones et al, 2014: https://doi.org/10.1002/2014GB004813

Kirchner et al., 2020: https://doi.org/10.1021/acs.est.9b07009

Geerts et al., 2025: https://doi.org/10.5194/bg-22-355-2025

Lehman & Bach, 2025: https://doi.org/10.1038/s41561-025-01644-0

---

## Author Comment (AC2)

**Reply to referee 2**

The paper is a useful summary of the chemistry and the applicability of accelerated weathering of limestone or buffered accelerated weathering of limestone, and it deserves publication. Minor comments below.

We would like to thank the reviewer for the positive feedback and the constructive comments. Below we provide a response to all their comments and suggestions, and indicate how we have altered the manuscript in response; our responses are in blue, altered text is in shaded in grey.

1. Lines 39-59. Please revise this section because it could lead to confusion among "enhanced weathering", "enhanced rock weathering" , "mineralization", and "carbonation" (in the case the mineral obtained is a carbonate mineral). The studies by Rau and Caldeira, 1999, Renforth and Kruger, 2013, Caserini et al., 2021, cited as "enhanced rock weathering" processes, could be better identified as accelerated weathering of limestone, to avoid confusion with enhanced weathering (that is a CDR approach that removes atmospheric carbon).

The start of this paragraph is rewritten to prevent confusion with enhanced weathering as CDR technology, Line 38 – 59:

"Industrial point-source $CO_2$ emissions from waste gas streams can be partially mitigated by geochemical-based processes in which $CO_2$ is reacted with solid carbonate or silicate rocks in the presence of water, which aims to enhance the natural weathering process of carbonate and silicate rocks (Rau and Caldeira, 1999; Renforth and Kruger, 2013; Caserini et al., 2021). This targeted weathering process can take place in situ, in which $CO_2$ is first captured from the flue gas and then injected into suitable silicate rock formations (basalts and ultramafic rocks). The $CO_2$ is then trapped by a carbonation reaction with the ambient silicate rock, thus ensuring a permanent, geological storage (Matter and Kelemen, 2009; Romanov et al., 2015; Gadikota, 2021; Cao et al., 2024). However, there are certain geomechanical risks associated with geological storage of $CO_2$, such as $CO_2$ leakage, induced seismicity, the loss of well integrity and surface uplift (Song et al., 2023). Moreover, suitable rock formations for storage are not always in close proximity to the $CO_2$-emitting installations, thus requiring compression and transport of $CO_2$.

Alternatively, the chemical weathering can also be executed under controlled conditions in a land-based reactor, close to the industrial point source. Mitigation of $CO_2$ emissions via such reactor-based methods can follow two main approaches, depending on whether silicates are used as feedstock material (usually referred to a "ex-situ mineral carbonation" technologies; Romanov et al., 2015; Gadikota, 2021, or "mineralization"; Campbell et al., 2022) or whether carbonates are used as weathering substrates (referred to a as "accelerated weathering of limestone"; Rau and Caldeira, 1999). In ex-situ mineral carbonation (ESMC), a finely-ground silicate mineral (e.g. olivine $Mg_2SiO_4$) is fed into a reactor, where it reacts at elevated temperature and pressure with $CO_2$ from a flue gas to eventually form stable carbonates (e.g. magnesite $Mg_2SiO_4$) - see recent reviews (Snæbjörnsdóttir et al., 2020; Veetil and Hitch, 2020; Thonemann et al., 2022). Alternatively, during the accelerated weathering of limestone (AWL), $CO_2$ is stripped from the flue gas using a mixture of seawater and limestone ($CaCO_3$) (Rau and Caldeira, 1999; Renforth and Henderson, 2017), and the resulting effluent is discharged into the sea."

2. Line 52-55 Please specify that what is called "ex situ mineral carbonation" (methods where alkaline minerals react with CO2, producing solid carbonate minerals) is also called "mineralization", as in Campbell et al (2022) https://doi.org/10.3389/fclim.2022.879133.

Mineralization is added as an alternative name for ex-situ mineral carbonation at Line 50 -53:

> "Mitigation of $CO_2$ emissions via such reactor-based methods can follow two main approaches, depending on whether silicates are used as feedstock material (usually referred to a "ex-situ mineral carbonation" technologies; Romanov et al., 2015; Gadikota, 2021, or "mineralization"; Campbell et al., 2022), …"

3. line 62: please specify that the CO2 removed by ocean alkalinization is atmospheric CO2

Adapted.

4. line 63. I don't see the need to add "chemical" between natural and weathering, since all the weathering processes are chemical processes.

'chemical' is removed.

5. Lines 91, 93, 99, and others: It is not clear what "upon discharge" means: just before the discharge of the process water or after the discharge? Sometimes, it seems just before (i.e.: … buffering with Ca(OH)2 upon discharge into the sea), in other cases, just after the discharge in seawater (upon re-exposure to atmospheric conditions, aqueous CO2 which is not stabilized by the increased AT will degas back to the atmosphere)

Clarified.

> "After discharge into the surface ocean, there is no longer any $CO_2$ transfer to the atmosphere"
>
> "The process water is discharged into the sea without any further treatment after which it re-equilibrates with the atmosphere at the lower $pCO_2$ ($pCO_2 \approx 0.00042$ atm), and the excess $CO_2$ (i.e., the part of DIC not stabilized by the increased alkalinity) will degas back to the atmosphere."
>
> "… (4a) the unbuffered or (4b) buffered process water after discharge into the surface ocean."

6. Line 99 "(4a-b) the unbuffered or buffered". Please clarify that 4a is unbuffered and 4b is buffered.

Adjusted according to the suggestion of the reviewer.

7. Lines 119-125 (table 1). It should be stated in the title what (1) (2) (3) (4a) and (4b) in the first column means. Since just before figure 1 there is (i) (ii) (iii) and (iv), there could be some misunderstanding.

The different states with number and explanation are now explicitly stated in the caption of Table 1.

> "Theoretical values for alkalinity ($A_T$), dissolved inorganic carbon (DIC), pH and calcite saturation state ($\Omega_{calc}$) in the four consecutive states of the example AWL reactor: (1) the process water that is used as intake (the process water was collected from an offshore station near the Hoping power plant and the inlet and outlet of the cooling water drainage of the Hoping power plant (Chou et al., 2015)) , (2) the process water with elevated DIC after $CO_2$ uptake, (3) the process water enriched in $A_T$ and DIC after $CaCO_3$ dissolution, (4a) the unbuffered or (4b) buffered process water upon discharge. $\Delta DIC_{seq}$ is the DIC that is added to the process water due to dissolution from the gas stream and $\Delta DIC_{carb}$ is the DIC added through the dissolution of $CaCO_3$. The $pCO_2$, $A_T$ and DIC values (indicated by #) are based on values measured in a two-step AWL bench-top reactor (Chou et al., 2015). The values of $A_T$, DIC, pH, and $\Omega_{calc}$ (indicated with *) are calculated using CRAN:AquaEnv (Hofmann et al., 2010) for seawater at a temperature of 15 °C and salinity of 35."

8. Line 124: the pH for 4a, unbuffered process water upon discharge, is 8.16, quite high, very close to the 8.27 for the buffered case. The pH is quite higher than in Caldeira and Rau 2000 https://doi.org/10.1029/1999GL002364. Please add some comments on this point.

The pH in Table 1 (4a) is calculated in R using the CRAN:AquaEnv package. The pH in state 4a is calculated based on the alkalinity content at the outlet of the reactor and the DIC content after full re-equilibration with atmospheric $pCO_2$.

In Caldeira and Rau (2000), the pH for the "degassed to seawater $\Omega_{calc}$" is based on the alkalinity at the outlet of the reactor and the $\Omega_{calc}$ of 4.14. In this case, the seawater is not fully equilibrated with the atmospheric $pCO_2$ (0.000402 atm) and the $fCO_2$ of the seawater is still at 0.014808 atm. If we calculate the pH for Caldeira and Rau (2000) using an outlet alkalinity of 14808 $\mu mol\,kg^{-1}$ and full equilibration with the atmospheric $pCO_2$, we get a pH of 8.5 due to the higher alkalinity compared to our example.

9. Lines145-149. Add more recent experimental studies:

Hartmann, J., Suitner, N., Lim, C., Schneider, J., Marín-Samper, L., Arístegui, J., Renforth, P., Taucher, J., & Riebesell, U. (2023). Stability of alkalinity in ocean alkalinity enhancement (OAE) approaches – consequences for durability of $CO_2$ storage. Biogeosciences, 20(4), 781–802. https://doi.org/10.5194/bg-20-781-2023

Moras, C. A., Bach, L. T., Cyronak, T., Joannes-Boyau, R., & Schulz, K. G. (2022). Ocean alkalinity enhancement – avoiding runaway $CaCO_3$ precipitation during quick and hydrated lime dissolution. Biogeosciences, 19(15), 3537–3557. https://doi.org/10.5194/bg-19-3537-2022

References included.

10. Lines 221-224. I would further clarify the reason behind the additional CO2 removal through liming. This represents a novelty of this study that was not addressed in Caserini et al. (2021), because buffered AWL is a carbon dioxide storage process. In contrast, ocean liming is a carbon dioxide removal process.

In this context, liming is meant as the addition of $Ca(OH)_2$ in the buffering reactor before discharge of the process water to the sea. The use of "liming" could indeed cause misunderstanding. It is changed to "buffering with $Ca(OH)_2$".

11. Line 258. I think the exponent of the unit of measurement is 3, not 2.

The exponent is changed to 3.

12. Line 427-428. I would provide more details about this calcination-free process as a method for $Ca(OH)_2$ recovery, since $Ca(OH)_2$ recovered from steel slag is obtained through calcination, then used in the steel industry, and ultimately ends up in the steel slag. Furthermore, I would elaborate on whether this process has other potential environmental side effects and provide more insights into its availability, as it depends on the residuals of an industrial process.

We do not think expanding on the process of forming $Ca(OH)_2$ fits within the scope of our paper, and would distract from the overall message.

13. Lines 461-462. Please provide a reference for the value of 1 ton of CO2 produced per ton of Ca(OH)2.

The value is changed to $1 - 1.8$ tonnes of $CO_2$ per tonne of $Ca(OH)_2$., and we included two new references:

Oates, 2008: ISBN: 978-3-527-61201-7

Simoni et al., 2022: https://doi.org/10.1016/j.rser.2022.112765

14. Lines 504-514. It's worth adding that the problems of trace metals or other pollutants are much lower if AWL or BAWL are used just for the storage of the CO2 produced by calcination, i.e. in the case of electric calcination

Included at Line 539 -541:

*"To potential negative effects from trace elements and other pollutants can be mitigated by using of relatively clean waste gas streams (such as from the combustion of natural gas or calcination of $CaCO_3$) in (B)AWL applications."*

15. Lines 515-521. Regarding potential impacts on marine biota, I would also cite the recent study by Sánchez et al. (2024).

Sánchez, N., Goldenberg, S. U., Brüggemann, D., Jaspers, C., Taucher, J., & Riebesell, U. (2024). Plankton food web structure and productivity under ocean alkalinity enhancement. Science Advances, 10(49), eado0264. https://doi.org/10.1126/sciadv.ado0264

Included.

---

## Author Response (AR2)

**Reply to referee 1**

We would like to thank the reviewer for their constructive and positive feedback on our manuscript. Below we provide a response to all their comments and suggestions, and indicate how we have altered the manuscript in response; our responses are in blue, altered text is in shaded in grey.

There have been at least 2 related reviews of this topic in recent months here: https://scijournals.onlinelibrary.wiley.com/doi/full/10.1002/ghg.2311, and a paper by Dong et al. Potential of CO2 Sequestration through Accelerated Weathering of Limestone on Ships that is in press at Science Advances. You can probably get a prerprint of the latter from Jess Adkins jess@caltech.edu

We thank the reviewer for bringing these recent papers to our attention.

Damu et al. (2024) is a research article which addresses the potential of construction grade limestone compared to lab grade limestone in a one-step AWL reactor using potable water. They discuss the  $CO_2$  capture efficiency and effluent  $A_T$  based of different liquid to gas ratios for this specific reactor design. They highlight that  $CaCO_3$  dissolution is the limiting step in the AWL process.

Dong et al. (2025) investigate the potential of AWL for ship-board applications both through benchtop-scale experiments and modelling. For there experiments they use a series of two-step AWL reactors changing water flow rate, solid holdup, and flow regime. They then modelled the instantaneous total efficiency and conversion efficiency for a ship-scale reactor while varying the solid holdup and limestone particle grainsize.

These two papers are experimental studies. Our paper provides a framework to compare the efficiencies of different reactor types based on inlet and outlet alkalinity and DIC values. We then provide an example of the use of this framework for different existing reactor designs. We further discuss the different reactor designs, the required feedstock for AWL, and the potential environmental impact. We have included the recommended papers in our discussion.

**My comments:**

1. Pg 1 line 19 Here and throughout - "large water usage" should read "large seawater usage" so as to make clear that the preferred application does not consume freshwater and that the real limitation is the cost and C footprint of pumping seawater, not the scarcity of seawater, unlike freshwater.

Recent studies, like Damu et al. (2024), use fresh water as feedstock for their AWL reactor However, in section 4: AWL feedstock, we highlight that seawater is the preferred feedstock for AWL.

2. Lines 64-73 Implies that rock weathering on land causes CO2 removal by the ocean. Rather, rock weathering on land consumes CO2 on land resulting in fully carbonated alkalinity that eventually reaches the ocean where it is stored. This alkalization of the ocean does not increase CO2 uptake by the ocean, but does increase the alkaline C stored there.

**We have adjusted the text to make the $CO_2$ uptake by $A_T$ increase more general.**

Line 64-73: The natural weathering of silicate and carbonate rocks generates  $A_T$  (Berner and Berner, 2004), which is defined as the excess of base (proton acceptors) over acid (proton donors) (Dickson, 1981; Zeebe and Wolf-Gladrow, 2001). Increasing the  $A_T$  content of the surface waters shifts the carbonate equilibrium away from dissolved  $CO_2$  towards bicarbonate ( $HCO_3^-$ ) and carbonate ( $CO_3^{2-}$ ) ions. As a result, the  $PCO_2$  of the surface water is reduced which drives a flux of  $CO_2$  from the atmosphere towards the surface water. This increases the amount of  $CO_2$  that can be sequestered and stored as dissolved inorganic carbon (DIC; defined as the sum of the aqueous  $[CO_2]$ ,  $[HCO_3^-]$ , and  $[CO_3^{2-}]$  concentrations; Zeebe and Wolf-Gladrow, 2001) in the ocean. This natural process of ocean alkalinization, induced by the chemical weathering of rocks, has regulated atmospheric  $CO_2$  and stabilized the climate over geological time scales (Berner et al., 1983).

**3. Line 87 Delete "out"**

"consisting out of ..." is changed to "consisting of ...".

4. Table 1. I find some inconsistencies in the measured versus calculated data presented. For example, taking the listed At and DIC reported by Chou et al for ambient seawater (state 1), I get a pCO2 of 666 uatms, pH=7.84 and an Omega(ca) = 2.48 using CO2SYS, in contrast to those in table 1 of 420, 7.94 and 2.50 respectively. For state 2, the table lists At= 2.26 and DIC=2.96, that according to my calcs should yield pCO2 = 0.19 atm, pH= 6.42 and Omega(ca)= 0.11, whereas Table 1 listed 0.15, 6.52 and 0.11, respectively. For State 3 the Chou et al data are listed as 2.64 and 3.15 for At and DIC, respectively. These values yield a calculated pCO2 of 0.14 atm, pH= 6.63 and Omega(ca) = 0.20, in contrast to 0.15 atm, 6.72 and 0.20 listed in table 1, respectively. For States 4a and b, my calcs based on listed At and DIC are pretty close to those listed. The CaCO3 saturation states seem pretty accurately represented, which is the critical thing, however, with the data clooged together like they are, some non-representative data can arise, in particular pCO2=420 vs 666 uatms for State 1 seawater.

The  $pCO_2$  represented in Table 1 is the ambient air or gas stream  $pCO_2$ . We agree that this is not well formulated in the text. We elaborated on this fact and added a column with the process water  $pCO_2$  ( $fCO_2$ ).

Line 115: "Table 1 shows the values for pCO2, gas, pCO2,water, AT, DIC, pH, and  $\Omega_{calc}$ ..."

The other slight differences between the results calculated with AquaEnv and CO2SYS could result from the difference in the used dissociation constants (k1k2, khf or kf, and khso4 or ks). In AquaEnv the default value of the first and second dissociation constant of carbonic acid (k1k2) is from Leuker et al. (2000), the HF dissociation constant (khf or kf) is from Dickson (1990), and the HSO4 dissociation constant (khso4 or ks) is from Dickson (1990).

Table 1: Differences between the calculated values for state 1 - 3 between AquaEnv and CO2SYS for a temperature of 15 °C and a salinity of 35.

|         |                        | AquaEnv |                           | CO2SYS                 |        |                           |  |
|---------|------------------------|---------|---------------------------|------------------------|--------|---------------------------|--|
|         | fCO 2 (atm) | pH (-)  | $\Omega_{ m calcite}$ (-) | fCO 2 (atm) | pH (-) | $\Omega_{ m calcite}$ (-) |  |
| State 1 | 0.000656               | 7.93    | 2.50                      | 0.000666               | 7.84   | 2.48                      |  |
| State 2 | 0.0189                 | 6.52    | 0.11                      | 0.19                   | 6.42   | 0.11                      |  |
| State 3 | 0.0139                 | 6.72    | 0.20                      | 0.14                   | 6.63   | 0.20                      |  |

5. Line 155 You mean "Because the input of AT from CaCO3 dissolution is twice that of the DIC supplied by CaCO3"? On the other hand, eq 1 show 1 mole of AT per mole of DIC produced, so what is meant here? There are 2 moles of potential alkalinity per mole of CaCO3.

With the sentence "... the input of  $A_T$  from  $CaCO_3$  dissolution is twice that of DIC ...", we mean that during  $CaCO_3$  dissolution  $A_T$  and DIC increases in a 2:1 ratio. We have elaborated on this in the manuscript.

Line 155: "Because the input of  $A_T$  from  $CaCO_3$  dissolution is twice that of DIC (2:1 ratio of  $A_T$  to DIC production), ..."

For Eq.  $1(CO_2 + H_2O + CaCO_3 \rightarrow Ca^{2+} + 2HCO_3^-)$ , in the manuscript, the logic we follow goes as follows.

On the left hand side  $(CO_2 + H_2O + CaCO_3)$ , there is already 1 mole of DIC (in the form of aqueous  $CO_2$ ) and 0 mole of  $A_T$  (as aqueous  $CO_2$  is not part of the  $A_T$  equation (Zeebe and Wolf-Gladrow, 2001)).

On the right hand side  $(Ca^{2+} + 2HCO_3^-)$ , there are 2 mole of DIC and 2 mole of  $A_T$  (2HCO3-, as  $HCO_3^-$  is both part of the DIC and  $A_T$  pool).

This results in a net increase of 1 mole of DIC and 2 mole of  $A_T$ .

Thus, during the dissolution of CaCO3, there is a 2:1 ratio of  $A_T$  to DIC that is produced.

6. Lines 165-6. "However, one can easily show that equilibration followed by mixing, provides the same CO2 transfer as mixing followed by equilibration." Not always true. If the mixing involves vertical mixing that could cause the CO2 supersaturated water to lose contact with the atmosphere, then full equilibration with air could take +1kyrs. This is touched on later in the paper when discussing efficiencies.

We have change the paragraph to account for the vertical mixing.

Line 163 - 166; "In our scheme, we assumed that the effluent process water first equilibrates with the ambient atmosphere, before it is mixed with the surrounding seawater. In reality, the process water will be mixed first with ambient seawater. If mixing involves vertical mixing of the process water supersaturated with  $CO_2$ , full equilibration will not be reached."

7. Line 173 it's rather than its.

The typo has been corrected.

8. Line 199-200 "DeltaDIC buf seq represents the DIC that is retained (i.e. prevented from efflux to the atmosphere) due to the Ca(OH)2 buffering of the effluent (in the unbuffered scenario DeltaDIC buf seq = 0)." Not clear/true. CaCO3 + CO2 also generates some buffered, alkaline C from CO2 as Ca++ 2(HCO3)- and CO3aq—that will not degass to the atmosphere. So, use of CaCO3 is also offers some buffered sequestration. Use of unbuffered here is not appropriate. Less vs more buffered?

In the manuscript, we separate between an unbuffered and a buffered AWL (Caserini et al., 2021) scenario. Throughout the text buffering refers to the addition of  $Ca(OH)_2$  to the effluent process water after the  $CaCO_3$  dissolution step.  $\Delta DIC_{seq}^{unbuf}$  present the amount of DIC that is sequestered from the gas stream in the unbuffered scenario (without addition of  $Ca(OH)_2$  to the effluent at the outlet of the reactor).  $\Delta DIC_{seq}^{buf}$  present the extra DIC that sequestered in the buffered scenario through the addition of  $Ca(OH)_2$  to the effluent after  $CaCO_3$  has taken place. A  $\Delta DIC_{seq}^{buf}$  of 0, does not mean that there is no DIC that is sequestered from the gas stream by  $CaCO_3$  dissolution but that there is no addition of  $Ca(OH)_2$ , and thus no additional buffering has taken place.

To make this point more clear, we have updated this section.

Line 197–201: "DICinlet is the DIC value measured in the process water at the inlet,  $\Delta DIC_{carb}$  denotes the DIC that originates from CaCO3 during dissolution,  $\Delta DIC_{seq}^{unbuf}$  represents the DIC in the process water that originates from net CO2 sequestration from the flue gas in the reactor through the increase in  $A_T$  from CaCO3 dissolution.  $\Delta DIC_{seq}^{buf}$  represents the DIC that is not

sequestered by  $CaCO_3$  dissolution that is retained (i.e. prevented from efflux to the atmosphere) due to the  $Ca(OH)_2$  addition to the effluent (in the unbuffered scenario  $\Delta DIC_{seq}^{buf} = 0$ ).

9. Eq 8 There is (still) a math operation symbol missing prior to the last term.

$$Eq. \ 8: \ DIC_{final} = f(A_{T,final}, pCO_{2,atm}) \approx DIC_{inlet} + \left(\frac{\partial DIC}{\partial A_T}\right)_{pCO_{2,atm}} A_{T,carb}$$

Is changed to  $DIC_{final} = f(A_{T,final}, pCO_{2,atm}) \approx DIC_{inlet} + (\partial DIC_{\partial A_T})_{pCO_{2,atm}} \Delta A_{T,carb}$ , to make the multiplication sign (. , as indicated by the red circle) more clear for readers.

10. Eq 9 is incorrect because it does not include a DelatDICbuffered component that is derived from CO2 via the reaction with CaCO3 and water. This eq assumes that there is zero buffered C storage derived from CO2 in reaction with CaCO3, which is false.

See response to comment 8 for explanation of the  $\Delta DIC_{seq}^{buf}$ .

Equation 9 includes the increase in DIC through  $A_T$  addition from CaCO3 dissolution and accounts for the re-equilibration with the atmosphere.

$$Eq. \ 9: \Delta DIC_{seq}^{unbuf} = DIC_{final} - DIC_{inlet} - \Delta DIC_{carb}$$

With
$$DIC_{final} = DIC_{inlet} + (\frac{\delta DIC}{\delta A_T})_{pCO_{2,atm}} * \Delta A_{T,carb}$$

Thus
$$\Delta DIC_{seq}^{unbuf} = (\frac{\delta DIC}{\delta A_T})_{pCO_{2,atm}} * \Delta A_{T,carb} - \Delta DIC_{carb}$$

The first term  $((\frac{\delta DIC}{\delta A_T})_{pCO_{2,atm}} * \Delta A_{T,carb})$  represents the extra DIC that can be stored in the process water due to the  $A_T$  increase from  $CaCO_3$  dissolution after re-equilibration with the atmosphere.

The second term ( $\Delta DIC_{carb}$ ) subtracts the DIC that is produced during the CaCO3 dissolution and that is thus not sequestered from the flue gas.

11. Lines 241-44 the total DIC increase in the equilibrated effluent water amounts to ΔDICtotal = 0.25mM in the unbuffered case, of which 76 % (0.19 mM) originates from CaCO3 dissolution and 24% (0.06 mM) is due to CO2 sequestration from the flue gas." Actually, 2.62 – 2.26 = 0.36 At produced = 0.18 mM C generated from the dissolution of CaCO3, but close enough? You imply that that 0.06mM is unbuffered dissolved CO2 when in fact most of it is Ca(HCO3)2aq + CaCO3aq, not CO2 or H2CO3!? A DeltaDIC/DelatAT = 0.25/0.36 = 0.69 seems very low. The assumption here that there is zero buffered CO2 storage is incorrect.

As can be seen from Table 1 in the manuscript,  $A_T$  in state 4a is 2.64 mM and not 2.62 mM as suggested by the calculation in the comment.

The change in  $A_T$  in the unbuffered scenario ( $A_T$  state  $4a - A_T$  state 1 = 2.64 - 2.26 = 0.38 mM) is 0.38 mM produced during CaCO3 dissolution.

$$Eq. \ 3: \Delta DIC_{total} = \Delta DIC_{seq}^{unbuf} + \Delta DIC_{seq}^{buf} + \Delta DIC_{carb}$$

 $\Delta DIC_{total} = DIC_{state4a} - DIC_{state1} = 2.38 \ mM - 2.13 \ mM = 0.25 \ mM$  (see Table 1 in the manuscript).

With  $\Delta DIC_{seq}^{buf} = 0$  mM for the unbuffered scenario

With  $\Delta DIC_{carb} = 0.19$  mM as for every mole of CaCO3 dissolution, 1 mole of DIC is produced and 2 mole of  $A_T$  (0.38/2 = 0.19).

Thus  $\Delta DIC_{seq}^{unbuf} = 0.06$  mM (see response to comment 8 for explanation of  $\Delta DIC_{seq}^{unbuf}$ )

The  $\frac{\delta DIC}{\delta A_T} = \frac{0.25}{0.38} \approx 0.65$  is indeed low. This can be explained by the fact that the inlet water used in Chou et al. (2015) was not in equilibrium with the atmosphere (fCO2 of 0.000656 atm) as pointed out by the reviewer. In state 4a, the process water is in equilibrium with the atmosphere (fCO2 is 0.000420). If the inlet process water would be in equilibrium with the atmosphere, the DIC content would be 2.06 mM (calculated with AquaEnv). This would result in a  $\Delta DIC_{total}$  of 0.32 mM and a  $\frac{\delta DIC}{\delta A_T} = \frac{0.32}{0.38} \approx 0.84$ .

For Table 1 and the example calculations based on Chou et al. (2015), the aim was to give an example of the different states for a representative real life (bench-top) reactor setup. All the values calculated in Table 1 and used in the example calculations are based on the measured inlet and outlet  $A_T$  and DIC from the two-step bench-top reactor from Chou et al. (2015).

We have highlighted the fact that the inlet process water is not at equilibrium with the atmosphere both in Table 1 and after the calculations.

Line 114-116: "Table 2 shows the values for  $pCO_{2, gas}$ ,  $pCO_{2, water}$ ,  $A_T$ , DIC, pH, and  $\Omega_{calc}$  in each of the four states for a representative case, which is based on data reported from a two-step bench-top reactor consisting of a separate gas-liquid and liquid-solid reactor (Chou et al., 2015, reactor design as further discussed below)."

| State       | pCO 2,gas (atm) | pCO 2,water (atm) | A T
(mM) | DIC
(mM) | ADIC seq (mM) | ADIC carb (mM) | рН
(-)  | $\Omega_{calc}$ |
|-------------|----------------------------|------------------------------|------------------------|-------------|--------------------------|---------------------------|------------|-----------------|
| (1)         | 0.000420                   | 0.000656*                    | 2.26#                  | 2.13#       | 0                        | 0                         | 7.93*      | 2.50*           |
| (2)         | 0.15 #                     | $0.0189^{*}$                 | 2.26                   | $2.96^{*}$  | 0.83                     | 0                         | $6.52^{*}$ | $0.110^{*}$     |
| (3)         | 0.15                       | $0.0139^{*}$                 | $2.64^{\#}$            | $3.15^{\#}$ | 0.83                     | 0.19                      | 6.72*      | $0.203^{*}$     |
| (4a) | 0.000420                   | 0.000420                     | 2.64                   | 2.38*       | 0.06                     | 0.19                      | 8.16*      | 4.62*           |
| (4b) | 0.000420                   | 0.000420                     | 3.56*                  | 3.15*       | 0.83                     | 0.19                      | 8.27*      | 7.74*           |

Line 251 - 252: It has to be noted that the inlet process water for this example from Chou et al. (2015) was not in equilibrium with the atmosphere (pCO2,water = 0.000656 atm instead of 0.000420 atm).

**12. Lines 247-8 If Ca(OH)2 is so great, why even bother with CaCO3?**

The calculation in this section are specific for the case of the bench-top two-step reactor of Chou et al. (2015). In line 149 to 152 in the manuscript, it is mentioned that the CaCO3 dissolution could still be significantly improved. The two recent papers mentioned by the reviewer both highlight that the CaCO3 dissolution is the limiting step in AWL.

It is thus not that Ca(OH)2 is so much better than CaCO3, but CaCO3 is the limiting step and needs to be further improved (e.g. by increasing the reaction time (Kirchner et al., 2020; Caserini et al., 2021), reducing particle size (Caserini et al., 2021; Kirchner et al., 2020), or increasing the hydrostatic pressure (Caserini et al., 2021)). The steps taken to improve CaCO3 dissolution is mentioned later in the manuscript under section 3: Different reactor designs for AWL.

Like proposed by Caserini et al. (2021),  $Ca(OH)_2$  can be used to buffer the excess unreacted  $CO_2$  but the use of  $Ca(OH)_2$  comes with a additional  $CO_2$  and energy penalty from the production process, which is mentioned in section 4: AWL feedstock.

Line 149 -152: "Note that the effluent at state 3 in the example two-step reactor is not in equilibrium with respect to  $CaCO_3$  dissolution ( $\Omega_{calc} < 1$ , Table 1). This indicates that the effectiveness of  $CaCO_3$  dissolution in the reactor design of Chou et al. (2015) could still be improved (e.g. by implementing a longer residence time)."

13. Line 281-2 You mean 150,000 m3. OK so this example is very water inefficient, but not representative of what an optimized system can do as later shown?

We have changed  $150.000 \text{ m}^3$  to  $150 000 \text{ m}^3$ .

We have removed "thus illustrating the large water footprint of AWL" as this indeed is only the case for this specific example and is not representative for more optimized reactor designs.

14. Line 318-21 What evidence is there that these numbers are representative of optimized AWL systems?

These values are indeed not representative for more optimized reactor designs with more efficient  $CaCO_3$  dissolution. We have added an extra sentence to elaborate on the fact that the difference in unbuffered (only  $CaCO_3$  dissolution) and buffered ( $CaCO_3$  dissolution and  $Ca(OH)_2$  addition)  $CO_2$  sequestration efficiency will become smaller.

Line 325-326: "However, when improving reactor designs to increase the CaCO3 dissolution efficiency the gap between the unbuffered and buffered CO2 sequestration efficiency will become smaller."

15. Line 440-2 Why not add the Ca(OH)2 at the beginning of the DR pipe rather than awkwardly at the end?

The idee of the dissolution reactor (DR) from Caserini et al. (2021), which is a tubular reactor that extends below the water surface to the deeper parts of the coastal zone to improve the  $CaCO_3$  dissolution reaction by increasing the hydrostatic pressure. If  $Ca(OH)_2$  would be added at the start of the DR,  $A_T$  would mainly be produced by  $Ca(OH)_2$  as the dissolution of  $Ca(OH)_2$  is faster than the dissolution of  $CaCO_3$ . This would increase the  $\Omega_{calcite}$  and reduce

the amount of CaCO3 that would eventually dissolve. Therefore, in the DR as much CaCO3 is dissolved as possible. Only the necessary amount of  $Ca(OH)_2$  is added after the DR in the buffering reactor (BR), which is much shorter than the DR, to compensate for the leftover unreacted  $CO_2$ .

16. Line 469 – Need to emphasize that AWL is not going to impact freshwater supply, but it will require lots of seawater where pumping cost will be the main issue and the need for coastally located CO2 sources.

See comment 1.

---

## Author Response (AR3)

**Reply to referee 1**

We would like to thank the reviewer for their constructive and positive feedback on our manuscript. Below we provide a response to all their comments and suggestions and indicate how we have altered the manuscript in response; our responses are in blue, altered text is in shaded in grey.

The term "water usage" and the idea that AWL has a large water usage needs to be change throughout the ms to "seawater usage". Without this clarification the average reader will infer that this can be freshwater usage and hence AWL will be limited by freshwater availability, competition with other freshwater demand and hence cost. Chai et al (2025, https://scijournals.onlinelibrary.wiley.com/doi/full/10.1002/ghg.2329) do describe freshwater AWL, but fail to consider the economics and practicality of this, esp in

freshwater AWL, but fail to consider the economics and practicality of this, esp in freshwater limited areas. Still, compare/contrast the experiments, modeling and conclusions of Chai et al. and those of this present ms? In any case, coastal AWL is not limited by the availability of seawater, but it can be limited by the cost of pumping that seawater to and from AWL reactors, hence the need to minimize seawater usage in order to reduce pumping cost, not because the resource is limited at coastal sites.

Recent studies, like Damu et al. (2024) and Chai et al. (2025), use fresh water for their AWL reactors and thus the term "water usage" over "seawater usage" is more accurate. The high water demand of the AWL process does restrict its possible usage to locations where the water source is essentially unlimited. In section 4: AWL feedstock, we highlight that seawater is the preferred feedstock for AWL applications.

2. Some of the dissolved carbon discharged by an AWL reactor will be in the form of excess, flue-gas-derived CO2 that has not or will not react with CaCO3 to from more stable Ca++ + HCO3-/CO3--, thus this CO2 fraction can easily degassed to air once discharged in the ocean. The question is are there cost effective ways to reduce this leakage and thus increase the fraction of CO2 removed from the flue gas that is "permanently" stored? While the addition of Ca(OH)2 to the effluent is a plausible solution, it has not been shown here that it is a practical and cost effective one, especially considering the current cost and CO2 footprint of its production. More cost-effective solutions could include site selection where the surface ocean water subduction rate is maximized thus limiting air/sea gas exchange, and/or where subsurface discharge of the effluent occurs (pg. 4). In any case, isn't the \$cost/t CO2 captured and stored what ultimately matters, not what max fraction of flue gas CO2 removal might be (theoretically) possible? The important questions not adequately addressed in the paper are can AWL be cost competitive with other forms of emissions reduction and what is the design that optimizes cost effectiveness?

The cost per tonne of  $CO_2$  sequestered will indeed determine the economic feasibility of the process, and this would require an assessment of different possibilities to optimize the  $CO_2$  sequestration efficiency and limit the  $CO_2$  degassing in cost-effective and environmentally safe ways. However, the goal of this manuscript is to provide a framework to compare the efficiencies of different reactor types based on measured inlet and outlet alkalinity and DIC values. An economical analysis of different reactor designs and different possible ways to reduce  $CO_2$  degassing would require a complete life-cycle analysis, which is well outside the scope of this paper.

3. Table 1 The pCO2 and DIC are uncharacteristically high for starting, ambient surface seawater and hence pH and Omega(ca) are uncharacteristically low. Implications for modeling results as used to infer realistic AWL performance?

We added a note in the manuscript text to highlight the fact that these values are uncharacteristically for ambient surface water conditions.

Line 118 – 119: "note that the inlet process water has a higher  $pCO_{2,water}$  and thus a higher DIC value than ambient surface water"

If we assume starting values of starting seawater equilibrated with atmospheric pCO2 and keep the alkalinity constant, the starting DIC changes from 2.13 mM to 2.06 mM. If we assume everything else would stay the same, the  $\Delta DIC_{seq}$  for the unbuffered and the buffered case become 0.13 and 0.90 mM respectively. This does not change the overall conclusion that the majority of the dissolved  $CO_2$  is degassed for this specific example.

4. Compare/contrast the experimental modeling output and conclusions here with those of Dong et al. (2025)? <a href="https://www.science.org/doi/10.1126/sciadv.adr7250">https://www.science.org/doi/10.1126/sciadv.adr7250</a>

The results of Dong et al. (2025) discussed in section 3.2: Two-step reactor. As no inlet and outlet DIC and alkalinity values are provide by Dong et al. (2025) further calculations of different efficiencies and comparisons with their model results can not be made.